# Design and *in silico* evaluation of an mRNA vaccine against HTLV-1 using AI-driven reverse vaccinology approaches

**Nadia Seifi**[1○], **Navid Nezafat**[2○], **Mohammad Soroosh Hajizade**[1], **Manica Negahdaripour**[1,2,3]*

**1** Department of Pharmaceutical Biotechnology, School of Pharmacy, Shiraz University of Medical Sciences, Shiraz, Iran, **2** Pharmaceutical Sciences Research Center, Shiraz University of Medical Sciences, Shiraz, Iran, **3** Department of Artificial Intelligence, School of Medicine, Shiraz University of Medical Sciences, Shiraz, Iran

○ These authors contributed equally to this work.
* negahdaripour@sums.ac.ir, manica.negahdaripour@gmail.com

## Abstract

Human T-lymphotropic virus type 1 (HTLV-1) is the first discovered human onco-genic retrovirus that can cause adult T-cell leukemia/lymphoma, HTLV-1-associated myelopathy/tropical spastic paraparesis, and several other diseases. Due to the poor prognosis of these diseases and the limited therapeutic modalities, the need for an HTLV-1 vaccine is crucial. The current study has used an artificial intelligence-driven reverse vaccinology approach to design an mRNA vaccine against HTLV-1. The two most antigenic proteins of the virus were selected and analyzed using multiple immunoinformatics tools to identify the antigenic immunodominant epitopes for T- and B-cells. Subsequently, the final selected epitopes and the adjuvant were con-nected using proper linkers. Subsequently, multiple 3D structures were modeled for the vaccine. After refining and evaluating the modeled structures, the best model was selected as the final candidate vaccine structure. The proposed mRNA structure is a potential vaccine with suitable immunological and physicochemical properties against HTLV-1. Docking and simulation analyses showed a proper interaction between the vaccine and the corresponding receptor of the employed adjuvant. However, addi-tional experimental studies are required to further confirm the vaccine's efficacy.

## Introduction

Human T-lymphotropic virus-1 (HTLV-1) is an important human oncogenic agent, first isolated from relatively mature T-cells of some American patients with human lymphoma and leukaemia [1]. Clinically, HTLV-1 is the most important virus of its family and was the first proven case of a cancer-causing pathogen [2]. The virus is a single-stranded and enveloped RNA retrovirus known as a human type-C RNA tumour virus [1,2]. Although approximately 95% of people infected with HTLV-1

**Data availability statement:** All the data used in the study are included and explained in the manuscript.

**Funding:** This work was supported by the Vice-Chancellor for Research, Shiraz University of Medical Sciences, Iran (Grant number: 28569).

**Competing interests:** The authors have declared that no competing interests exist.

remain asymptomatic, about 5% develop HTLV-1–associated diseases. HTLV-1 is generally linked to two diseases: HTLV-1-associated myelopathy/tropical spastic paraparesis (HAM/TSP) in roughly 2% and adult T-cell lymphoma (ATLL) in almost 4% of the infected people [2]. HTLV-1 is also associated with a broader spectrum of clinical conditions, including HTLV-1-associated uveitis, infectious dermatitis (IDH), large granular lymphocytic (LGL) leukaemia, and inflammatory pulmonary disorders such as alveolitis and pneumonitis. These reflect the virus's ability to induce chronic immune activation, lymphoproliferation, and tissue-specific inflammation [3].

According to the latest World Health Organization (WHO) estimation, 5–10 million people live with this infection, with approximately half residing in equatorial Africa. Endemic hotspots of the virus occur across all continents. High prevalence of HTLV-1 infection has been reported in Latin America and the Caribbean, among Indigenous Australians, in equatorial Africa, northeastern Iran, and southwestern Japan [4,5].

HTLV-1 is transmitted mainly through cell-containing bodily fluids, most notably breast milk, blood, and semen, with infection occurring predominantly via direct cell-to-cell contact [4]. The virus mainly targets CD4 + T lymphocytes, where it persists with low replication, promoting cellular proliferation, genetic instability, and chronic inflammation [6,7]. Viral entry is mediated by the SU and TM envelope glycoproteins [8,9]. HTLV-1 integrase facilitates stable proviral integration into host chromosomes, a key step in lifelong persistence. Integrase strand transfer inhibitors (INSTIs) originally developed for HIV-1, such as raltegravir and dolutegravir, have shown inhibitory activity against HTLV-1 *in vitro* [10].

Following integration, the provirus is maintained by clonal expansion with expression regulated by Tax and Rex viral proteins. Although HTLV-1 is non-cytopathic, regulatory factors such as Tax and HBZ modulate host transcriptional pathways, driving oncogenesis, immune dysregulation, and chronic inflammation [7]. Clinically, HTLV-1 is most strongly associated with ATL and HAM/TSP, with disease development influenced by transmission route, host genetics, including HLA type, and viral genetic variability [7,11].

Currently, clinical disease management focuses on monitoring and treating HTLV-1-associated conditions such as ATLL and HAM/TSP, as well as performing routine screening for comorbidities and potential co-infections. No treatment is suggested for asymptomatic infected people [4]. Given the lack of potential vaccine against this virus, developing an effective vaccine is urgent.

As a novel vaccine class, mRNA vaccines represent a promising prophylactic strategy against viruses [12]. They contain an mRNA sequence encoding a viral protein, usually from the viral outer membrane. Upon uptake of the mRNA vaccine, cells can produce the viral protein. Naturally, the immune system identifies the foreign protein and produces antibodies. Therefore, these vaccines can induce a balanced immune response, including both humoral and cellular immunity [13]. The benefits of mRNA vaccines include safety, effectiveness, versatility, and fairly low adverse reactions. Their industrial production is also simple, rapid, and relatively low-cost. These advantages make them a powerful platform and a rapidly growing field. However, they suffer from instability against temperature changes, high immunogenicity, population

variations, and the need for high-quality and standardized raw materials, plus specialized facilities for their synthesis, purification, and formulation [13,14].

Several HTLV-1 vaccine approaches (recombinant vaccinia virus vector, protein, and DNA vector vaccines) have been explored in recent years, yet no HTLV-1 vaccines are currently in clinical trials [15]. Multiple challenges must be addressed to induce an effective immune response with such vaccines. For example, the cell-to-cell transmission of HTLV-1 can happen through forming virological synapses (VS), cellular conduits, or extracellular viral assemblies. Another obstacle is the lack of detailed studies on the HTLV-1 Env structure. Env is a precursor protein consisting of gp46 and gp21, cleaved in the Golgi apparatus before transport to the cell surface. While the structure of gp21 is known, the structure of the gp46 subunit has yet to be determined [12].

Using bioinformatic approaches for rational vaccine design based on the genome data, with no need to culture the specific microorganisms, is called "reverse vaccinology" [13,14]. Nowadays, applying these approaches through machine learning (ML)- and artificial intelligence (AI)-driven tools plays a significant role in vaccine development [15]. Generally, computational tools help researchers evaluate and improve the essential characteristics of the studied vaccine [16]. Using AI-driven algorithms and *in silico* tools results in a significant reduction of production time and cost due to the more effective selection of antigens, epitopes, linkers, adjuvants, and other components of the vaccine [17].

Following a reverse vaccinology approach, this project aimed to design an mRNA-based vaccine against HTLV-1 using bioinformatics and computational techniques, including AI- and ML-based algorithms.

## Methods

### Retrieval of viral protein sequences

The amino acid sequences of HTLV-1 target proteins (Taxonomy ID: 11908 NCBI), including 1) TAX, 2) HBZ, 3) gp21, 4) Gag (p15, p19, p24), 5) gp62, 6) p27, 7) pol, and 8) gag-pro-pol, were retrieved in FASTA format from the UniProt database (UniProtKB) (https://www.uniprot.org/uniprotkb) [18].

### Analysis of the allergenicity and antigenicity of selected proteins

The antigenicity of the eight above-mentioned proteins of HTLV-1 was evaluated by both ANTIGENpro (http://scratch.proteomics.ics.uci.edu/) and VaxiJen v2.0 (http://www.ddg-pharmfac.net/vaxijen/VaxiJen/VaxiJen.html) servers [19]. Then, their allergenicity was checked by AllerTOP v2.0 (https://www.ddg-pharmfac.net/AllerTOP/) [20] and AllergenFP v.1.0 (https://ddg-pharmfac.net/AllergenFP/) [21].

### Epitope selection

**Prediction of MHC-I- and MHC-II-binding and CTL epitopes.** CTLpred (https://webs.iiitd.edu.in/raghava/ctlpred/) was used to predict cytotoxic T-cell epitopes with artificial neural network (ANN)-based approaches, using the server's default cut-off scores: 0.51 ANN and 0.36 support vector machine (SVM) [22].

The IEDB (https://www.iedb.org/) [23] and Rankpep (http://imed.med.ucm.es/Tools/rankpep.html) [24] servers were used to predict 9-mer CTL epitopes for three MHC class I supertypes, namely A*01:01, A*02:01, and A*03:01 [25], and three MHC class II supertypes, including DRB1*01:01, DRB1*03:01, and DRB1*04:01 [26].

The Rankpep server (http://imed.med.ucm.es/Tools/rankpep.html) predicts peptides that bind to MHC-I and MHC-II molecules based on protein sequence/s or sequence alignments using position-specific scoring matrices (PSSMs). The NetMHCpan 4.1 EL prediction method (recommended epitope predictor) was used in IEDB.

**Prediction of linear B-cell epitopes.** BCPREDS (http://ailab-projects2.ist.psu.edu/bcpred/predict.html) [27] and the BepiPred-2.0 predictor on the IEDB server were used to determine the linear B-cell epitopes of the selected protein sequences [28].

**Final selection of immunodominant epitope regions.** After defining each of the epitopes (from all three types of epitopes) and a precise evaluation of overlaps between the predicted antigens, multiple epitope-rich regions were selected on two viral proteins, gp46 and pol. From here onwards, the selected epitope-rich segments will be called overlapping epitopes.

## Toxicity, allergenicity, and antigenicity evaluation of the selected epitopes

The selected overlapping epitopes in the previous step were evaluated by AllergenFP v.1.0, AllerTOP, Vaxijen v2.0, and ToxinPred2 (https://webs.iiitd.edu.in/raghava/toxinpred2/batch.html) [29].

In Vaxijen v2.0, the default threshold (0.4) was employed. Moreover, the default threshold (0.6) of the ToxinPred2 server was used.

## Construction of a multi-epitope potential vaccine candidate

The selected overlapping epitopes were linked to make six fusion polypeptide constructs using GPGPG [30], KK [31], and AAY [32] peptide linkers. The selected adjuvants, including PADRE [33,34] and three TLR4 agonists (mycobacterial heparin-binding hemagglutinin (HBHA) conserved [32], RS01, and RS09 – lipopolysaccharide mimotope synthetic peptides-[35]) were linked to other parts using EAAAK rigid linkers [31]. Fig 1 represents the schematic structure of the six designed combinations. The difference between the six constructs was the employed adjuvant; each protein construct encompassed two of the three adjuvants (RS01, RS09, and HBHA conserved).

## Prediction, validation, and refinement of 3D structures

The 3D structure of the designed constructs was predicted by three tools, including I-TASSER (https://zhanggroup.org/I-TASSER/) (Iterative Threading ASSEmbly Refinement) homology modeling tool [36], GalaxyTBM service of GalaxyWEB server (https://galaxy.seoklab.org/cgi-bin/submit.cgi?type=TBM) [37], and trRosetta server (https://yanglab.qd.sdu.edu.cn/trRosetta/) [38]. The I-TASSER server applies several steps to predict three-dimensional structures, including identifying template proteins with a structure similar to the investigated protein sequence, structural assembly, selection and modification of models, and finally, functional interpretation based on the structure using the COACH method. I-TASSER predicts a C-score for all five structures and a TM-score and RMSD for the first model to estimate the quality and accuracy of the predicted models [36].

GalaxyTBM introduces a new approach to template-based modeling (TBM). This method initially constructs the more reliable core region using multiple templates. Subsequently, it detects and remodels less reliable and variable areas, such as loops or terminals, using an *ab initio* technique [37].

trRosetta uses a deep neural network to predict the inter-residue geometries, which include orientations and distances. The forecasted geometries are subsequently utilized as restraints to lead the structure prediction process based on direct energy minimization. According to the reliability of the top forecasted distance and the convergence of the best models, trRosetta calculates the TM-score of the predicted model to estimate the accuracy of a given topology [38].

The 3D structures were evaluated using ProSA-web [39], and two tools of the SAVESv6.0 server (https://saves.mbi.ucla.edu/), namely PROCHECK [40] and ERRAT [41].

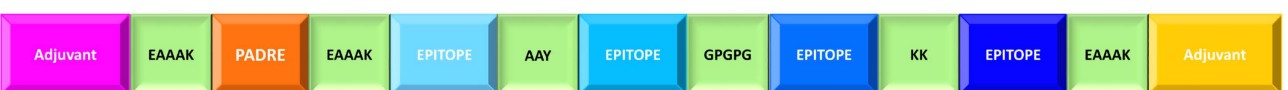

**Fig 1. Schematic representation of the different segments employed in the designed polypeptide vaccine.** The six designed constructs varied in the used adjuvants; each protein construct encompassed two of the three selected adjuvants (RS01, RS09, HBHA conserved).

ERRAT evaluates the non-covalent interactions between atoms [41]. The PROCHECK server models the stereochemical and geometrical constraints of the protein's 3D structure in the Ramachandran Plot [40].

The ProSA-web server (https://prosa.services.came.sbg.ac.at/prosa.php) calculates the overall protein quality score (z-score) obtained by comparing the 3D model of the target protein with similar proteins of the same size. The reference proteins' z-scores belong to those studied by X-ray crystallography or NMR spectroscopy.

During the assessment process, the 3D structures were visualized via PyMOL software. The best 3D structure was selected based on the results obtained from the mentioned servers.

The selected structure in the previous step was refined by GalaxyRefine2 (https://galaxy.seoklab.org/cgi-bin/submit.cgi?type=REFINE2) [42] services from the GalaxyWEB server.

GalaxyRefine2 conducts iterative optimization with various geometric operators to enhance the accuracy of the primary model [42].

After refinement, the obtained 3D structures were re-evaluated with ProSA-web, PROCHECK, and ERRAT servers, and the best-refined structure was selected for the next steps.

### Disulfide engineering of the construct

To enhance the construct's stability, disulfide engineering was conducted to identify residue pairs with the potential for cysteine mutation and disulfide bond formation. The 3D model of the vaccine protein (in PDB format) was submitted to the Disulfide by Design 2.0 (DbD2) server (http://cptweb.cpt.wayne.edu/DbD2/), and the analysis was performed using the program's default parameters [43].

### Physicochemical and stability analyses of the vaccine construct

To confirm the stability and the physicochemical properties of the final vaccine, SCooP (http://babylone.3bio.ulb.ac.be/SCooP/index.php) [44] and ProtParam (https://web.expasy.org/protparam/) [45] servers were used, respectively.

The SCooP server is assumed to follow a two-state folding transition. This server predicts a temperature-dependent stability curve for the protein based on its 3D structure [44].

The ProtParam server was used to compute different properties that can be deduced from the protein's sequence, such as the number of amino acids, molecular weight, formula, the total number of atoms, estimated half-life, aliphatic index, grand average of hydropathicity, theoretical pI, the total number of negatively charged residues, and the total number of positively charged residues [45].

In addition, a structural analysis was conducted to evaluate the effects of specific outlier amino acid residues on the overall protein conformation. The Cupsat server (Cologne University Protein Stability Analysis Tool) (https://cupsat.brenda-enzymes.org/) was utilized to model and analyze the structural impact of mutating these residues, providing insights into possible conformational changes. This program predicts ΔΔG, the difference in unfolding free energy between original and mutant proteins, using structural torsion angle and environment-specific atom potentials [46].

To evaluate the solubility of the vaccine construct, the Aggrescan3D server (https://biocomp.chem.uw.edu.pl/A3D2/) was utilized. This is a structure-based aggregation prediction tool that analyzes the aggregation properties of globular protein surfaces. The server provides a total score (derived from the sum of A3D scores across all residues) and an average score (obtained by normalizing the total score to the number of amino acids). More negative values of these scores are indicative of greater predicted solubility [47].

### Conformational B-cell epitope prediction

Conformational B-cell epitopes are discontinuous and are composed of patches of atoms exposed to solvent, formed by residues that do not have to be sequential.

The prediction of conformational B-cell epitopes was made after determining the final 3D structure of the vaccine using the ElliPro tool (http://tools.iedb.org/ellipro/) [48]. Default parameters, including a minimum score of 0.5 and a maximum distance of 6 Å, were used. This tool uses Thornton's method, a residue clustering algorithm, the Jmol viewer, and the MODELLER program to predict and visualize epitopes in a given protein. ElliPro predicts discontinuous (structural) epitopes based on the 3D structure of the input protein.

## Molecular docking to analyze the peptide vaccine interaction with TLR4

Molecular docking was performed between the final 3D structure of the vaccine and TLR4/MD2 receptor by ClusPro (https://cluspro.bu.edu/) [49]. The PDBsum server (https://www.ebi.ac.uk/thornton-srv/databases/pdbsum/) [50] and PPCheck server (https://caps.ncbs.res.in/ppcheck/) [51] were utilized to map the residue interaction between TLR4/MD2 and the vaccine.

The PROtein binDIng enerGY prediction (PRODIGY) tool (https://wenmr.science.uu.nl/prodigy/) of the HADDOCK server [52] was used to calculate the dissociation constant ($K_D$) and binding free energy ($\Delta G$) based on their 3D structure. The best-docked model was selected for further investigations through molecular dynamics (MD) studies.

## MD simulation studies

To stimulate the dynamics of the protein complex, which includes the vaccine and the TLR4/MD-2 complex, a molecular dynamics (MD) simulation was performed. The topology files were generated using the AMBER99SB-ILDN force field. The system was solvated with the SPC216 water model, and an electrolyte concentration of 0.15 M NaCl was added to replicate physiological conditions. The simulations were conducted using GROMACS version 2024 [53]. The system was placed in a cubic box under periodic boundary conditions, ensuring a minimum distance of 1.0 nm between the protein and the box walls. The LINCS algorithm was used to constrain all covalent bonds involving hydrogen atoms. Both NVT (constant number of particles, volume, and temperature) and NPT (constant number of particles, pressure, and temperature) ensembles were generated and equilibrated at 300 K and 1 bar, using the V-rescale thermostat and Parrinello-Rahman barostat, respectively. Before the production run, energy minimization was performed using the steepest descent algorithm. The production MD simulation was maintained for 100 ns, and subsequent analyses were conducted on the resulting trajectory.

To assess the stability and flexibility of the complex, several analyses were performed, including root-mean-square deviation (RMSD), root-mean-square fluctuation (RMSF), radius of gyration (Rg), and hydrogen bond assessments. The GROMOS clustering algorithm was applied to the trajectory to reduce the large number of frames into a representative set, with clusters identified based on a cut-off of 0.15 nm. The structure with the most neighbours was selected and removed from further clustering. Hydrogen bond interactions were analyzed using the H-bond module in GROMACS, applying the default criterion of a distance ≤3.5 Å. Data visualization and plot generation were conducted using QtGrace and Microsoft Excel 2019, while interaction diagrams were created with UCSF Chimera and PyMol 2.6 software.

## Prediction of population coverage

Population coverage prediction was conducted by the population coverage tool on the IEDB server (http://tools.iedb.org/population/) [54]. This server determines the effectiveness of each overlapping epitope worldwide or in a specific population or area by evaluating the prevalence of MHC-I and -II alleles associated with every overlapping epitope. The server provided the results by receiving the sequence of each overlapping epitope and the related MHC-I and -II [17].

## *In silico* immune simulation

The C-ImmSim server (https://kraken.iac.rm.cnr.it/C-IMMSIM/) [55] was used to simulate the vaccine's real-life immune response induction. The C-ImmSim server uses a PSSM derived from ML techniques. For immune simulation, three injections containing 1000 antigen proteins without LPS were set at the time steps of 1, 84, and 168. Since each time step

equals eight hours, each injection was 28 days (four weeks) apart from the previous one. The simulation steps were set at 1050, and the simulation volume (10), random seed (12345), and other parameters remained at default.

### Codon optimization

Codon optimization was performed using the GenSmart Codon Optimization Tool server (https://www.genscript.com/tools/gensmart-codon-optimization) for human as the expression host organism to improve transcription from mRNA vaccine and recombinant protein production later in the body.

The optimized DNA sequence was then reverse-transcribed into RNA using the DNA>RNA>Protein tool (http://biomodel.uah.es/en/lab/cybertory/analysis/trans.htm) and utilized in the mRNA construct as part of the open reading frame (ORF).

### Construction of the mRNA construct

After reverse translation of the designed protein and adding several segments to increase the efficiency and translation of mRNA, the mRNA vaccine construct was designed.

The tissue plasminogen activator (tPA) secretory signal and MHC I-targeting domain (MITD) sequences were obtained from the UniProt server with IDs P00750 and Q8WV92, respectively. Using an AAY linker, these two sequences were connected to the open reading frame ORF's 5' and 3' terminals, respectively.

Human β-globin was used for 5' UTR, and rabbit β-globin gene was used for 3' UTR, obtained from the NCBI server with ID NM_000518.5 and the GenBank server with ID V00882.1, respectively; and to improve stability, flanked the 5' and 3' ends. A 7-methyl(3-O-methyl) GpppG cap (ARCA) caps the 5' end, and a poly (A) tail with a length of 120 nucleotides was also added to the 3' end.

### RNA vaccine secondary structure prediction

Subsequently, the RNAfold tool of the ViennaRNA Package 2.0 (http://rna.tbi.univie.ac.at/cgi-bin/RNAWebSuite/RNAfold.cgi) [56] was employed to predict the mRNA vaccine's secondary structure. This server predicts the minimum free energy score (MFE) using the algorithm introduced by Zuker and Stiegler [56].

### Post-translational modification (PTM)

Post-translational modification (PTM) analyses of the vaccine were performed by NetNGlyc 1.0 (https://services.healthtech.dtu.dk/services/NetNGlyc-1.0/) [57], NetPhos 3.1 (https://services.healthtech.dtu.dk/services/NetPhos-3.1/) [58], and NetAcet 1.0 (https://services.healthtech.dtu.dk/services/NetAcet-1.0/) [59] servers, which identify glycosylation, phosphorylation, and acetylation sites, respectively.

For lipid PTMs, big-PI/GPI animals (https://mendel.imp.ac.at/gpi/gpi_server.html) [60] server and MyrPS/NMT server (https://mendel.imp.ac.at/myristate/) [61] were used for GPI modification and N-terminal glycine myristoyl, respectively.

## Results

### Retrieval of viral protein sequences

The eight protein sequences of the HTLV-1 virus were retrieved from the UniProt database. Table 1 contains the accession number and length of each protein, and their sequences are presented in S1 Table.

### Selection of the best antigenic proteins and analysis of allergenicity and antigenicity of the selected antigens

Based on the results obtained from the ANTIGENpro and VaxiJen v2.0 servers (Table 2), two antigens, gp62 and pol, were selected for further analysis. According to AllerTOP v. 2.0 and AllergenFP v.1.0, both proteins were "probable non-allergens".

**Table 1. The length and accession number of HTLV-1 proteins.**

| Antigen | Length | Primary accession number |
|---|---|---|
| TAX | 353 | P03409 |
| HBZ | 209 | P0C746 |
| gp21 | 177 | A0A023HB55 |
| Gag (p15, p19, p24) | 429 | P03345 |
| gp62 | 488 | P23064 |
| p27 | 274 | Q9QR98 |
| pol | 731 | Q82324 |
| gag-pro-pol | 1462 | P03362 |

**Table 2. Antigenicity of HTLV-1 different proteins.**

| Antigen | VaxiJen (overall prediction for the protective antigen) | ANTIGENpro |
|---|---|---|
| TAX | 0.4610 (Probable ANTIGEN) | 0.793235 |
| HBZ | 0.3563 (Probable NON-ANTIGEN) | 0.795961 |
| gp21 | 0.5779 (Probable ANTIGEN) | 0.395082 |
| Gag (p15, p19, p24) | 0.3884 (Probable NON-ANTIGEN) | 0.884140 |
| gp62 | 0.5496 (Probable ANTIGEN) | 0.863476 |
| p27 | 0.3904 (Probable NON-ANTIGEN) | 0.855717 |
| pol | 0.4143 (Probable ANTIGEN) | 0.756956 |
| gag-pro-pol | 0.4511 (Probable ANTIGEN) | 0.648948 |

## Immunodominant epitope selection

Finally, five regions were selected after predicting MHC-I and MHC-II binding epitopes, CTL epitopes, and linear B-cell epitopes. These selected regions cover all four types of the mentioned epitopes (Table 3).

## Toxicity, allergenicity, and antigenicity assay of the selected overlapping epitopes

The five selected overlapping epitopes were evaluated by ToxinPred2, Vaxijen v2.0, and AllergenFP v.1.0 to assess their toxicity, antigenicity, and allergenicity, respectively. Based on the results of the conducted assessments shown in Table 4, the four best overlapping epitopes, including gp62−1, gp62−3, Pol-1, and Pol-2, were selected for designing the vaccine.

**Table 3. The final regions of gp62 and Pol proteins selected for vaccine construction.**

| The overlapping epitope's assigned name | Start-stop amino acids | Selected sequence |
|---|---|---|
| gp62−1 | 50-91 | WTLDLLALSADQALQPPCPNLVSYSSYHATYSLYLFPHWIKK |
| gp62−2 | 233-290 | SLSTWHVLYSPNVSVPSSSSTPLLYPSLALPAPHLTLPFNWTHCFDPQIQAIVSSPCH |
| gp62−3 | 336-416 | GSMSLASGKSLLHEVDKDISQLTQAIVKNHKNLLKIAQYAAQNRRGLDLLFWEQGGLCKALQEQCCFLNITN-SHVSILQER |
| Pol-1 | 357-407 | WRCLNIFLDSKYLYHYLRTLALGTFQGRSSQAPFQALLPRLLSRKVVYLHH |
| Pol-2 | 522-580 | AISATQKRKETSSEAISSLLQAIAYLGKPSYINTDNGPAYISQDFLNMCTSLAIRHTTH |

**Table 4. Toxicity, allergenicity, and antigenicity of selected overlapping epitopes from gp62 and Pol antigens.**

| Antigen | ToxinPred | Vaxijen v2.0 (overall prediction for the protective antigen) | AllergenFP v.1.0 |
|---------|-----------|--------------------------------------------------------------|------------------|
| gp62−1 | Non-Toxin | 0.5763 (Probable ANTIGEN). | PROBABLE NON-ALLERGEN |
| gp62−2 | Non-Toxin | 0.3830 (Probable NON-ANTIGEN). | PROBABLE ALLERGEN |
| gp62−3 | Non-Toxin | 0.5197 (Probable ANTIGEN) | PROBABLE NON-ALLERGEN |
| Pol-1 | Non-Toxin | 0.5273 (Probable ANTIGEN) | PROBABLE NON-ALLERGEN |
| Pol-2 | Non-Toxin | 0.6237 (Probable ANTIGEN) | PROBABLE NON-ALLERGEN |

## Construction of the different arrangements of the protein vaccine

Six different constructs were designed based on the previously mentioned methods by permutating overlapping epitopes and adjuvants in various orders. These primary protein constructs are shown in S2 Table.

## Validation of the designed protein constructs

After evaluating the designed constructs for allergenicity, antigenicity, and toxicity using AllerTOP v.2, AllergenFP v.1.0, Vaxijen v2.0, and ToxinPred (results in Table 5), three constructs were selected for further analyses.

## Prediction, refinement, and validation of the 3D modelled structures

For each selected construct, five 3D structures were proposed by the I-TASSER and GalaxyWEB server and one by the trRosetta server, making a total of eleven 3D structures per construct.

Each 3D structure was evaluated individually using ProSA-web and two tools of the SAVESv6.0 server, namely PRO-CHECK and ERRAT. By reviewing the evaluation results of the mentioned servers (S3 Table), the best 3D structure was selected and visualized by PyMOL software.

The selected 3D structure was refined by the GalaxyRefine2 service, which gives ten refined conformational structures for each submitted 3D structure.

**Table 5. Allergenicity, antigenicity, and toxicity validation of six designed constructs using AllerTOP v.2, AllergenFP v.1.0, Vaxijen v2.0, and ToxinPred servers.**

| Construct's name* | ToxinPred | AllerTOP v.2 | Vaxijen v2.0 (Overall prediction for protective antigens) | AllergenFP v.1.0 |
|-------------------|-----------|--------------|-----------------------------------------------------------|------------------|
| Construct 1 | Non-toxin | Probable allergen | 0.5253 Probable antigen | Probable non-allergen |
| Construct 2 | Non-toxin | Probable allergen | 0.5291 Probable antigen | Probable non-allergen |
| Construct 3 | Non-toxin | Probable allergen | 0.5195 Probable antigen | Probable non-allergen |
| Construct 4 | Non-toxin | Probable non-allergen | 0.5178 Probable antigen | Probable non-allergen |
| Construct 5 | Non-toxin | Probable non-allergen | 0.5576 Probable antigen | Probable non-allergen |
| Construct 6 | Non-toxin | Probable non-allergen | 0.5218 Probable antigen | Probable non-allergen |

* The full sequence of each construct is shown in S1 Table.

The refined 3D structures in the previous step were re-evaluated with ProSA-web, PROCHECK, and ERRAT servers (results in S4 Table), and one final structure was selected (Fig 2). The z-score and the overall quality factor of the selected final 3D structure were −7.59 and 95.6679, calculated by ProSA-web and ERRAT servers, respectively. The obtained Ramachandran plot also indicates 92.3% of the residues are in the most-favored regions, and 5.8%, 1.2%, and 0.8% of

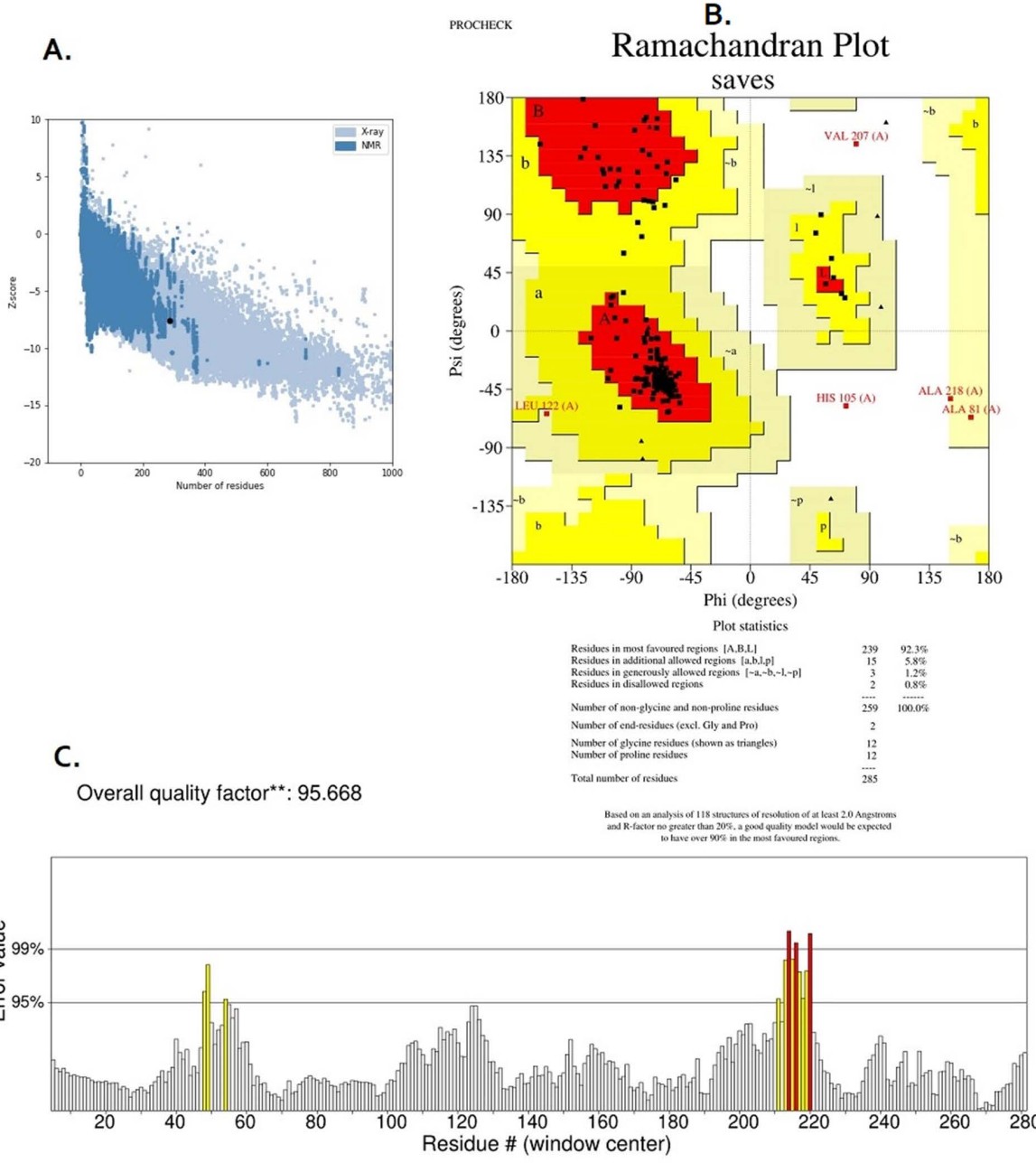

**Fig 2. Validation results of the final refined 3D model of the vaccine.** (A) The ProSA-web server results. The horizontal and vertical axes represent the number of residues and the overall protein quality score (z-score), respectively. The z-score is obtained by comparing the 3D model of the target protein with similar proteins of the same size. The calculated z-score for the selected structure was −7.59, shown here by the black dot. (B) Ramachandran plot analyzed by the PROCHECK server. (C) ERRAT server result. The horizontal axis represents the amino acid residues of the protein, and the vertical axis represents the error value. Regions of structure (amino acids) that can be rejected at the 95% and 99% confidence level are shown in yellow and red, respectively.

the residues are in additionally-allowed, generously-allowed, and disallowed regions, respectively. Fig 3 shows the final refined 3D structure.

## Evaluation of stability and physicochemical parameters

The stability and physicochemical properties of the vaccine were checked by SCooP and ProtParam. The SCooP server showed the stability of the vaccine 3D structure at −30 to +60 °C (Fig 4). Table 6 presents the physicochemical properties of the sequences, predicted by ProtParam.

The impact of mutations at histidine 105 and valine 207, identified as outlier residues in the Ramachandran graphs, was evaluated by replacing each with 19 other amino acids to assess their effect on the vaccine structure. The results of these analyses are presented in S5 Table, respectively, for histidine at position 105 and valine at position 207. These tables highlight the structural deviations compared to the original conformation.

Analysis via the DbD2 server identified several candidate residue pairs for disulfide bond formation, as shown in Table 7.

According to the results obtained from the Aggrescan3D server, the designed vaccine construct is likely to possess favorable solubility characteristics, with a total score value of −205.1575 and an average score of −0.7189.

## Conformational B-cell epitope prediction

Nine conformational B-cell epitope regions were predicted on the vaccine's selected 3D structure using ElliPro (Table 8 and S1 Fig).

## Molecular docking of the peptide vaccine and TLR4/MD2

The molecular docking between the vaccine and TLR4/MD2, conducted by ClusPro, resulted in 30 balanced models and their binding energy scores.

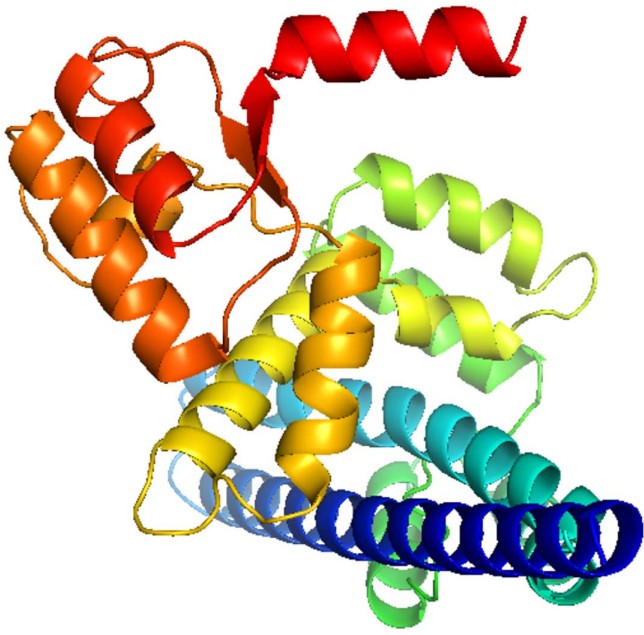

**Fig 3. The final 3D model of the designed protein vaccine, visualized by PyMOL software.**

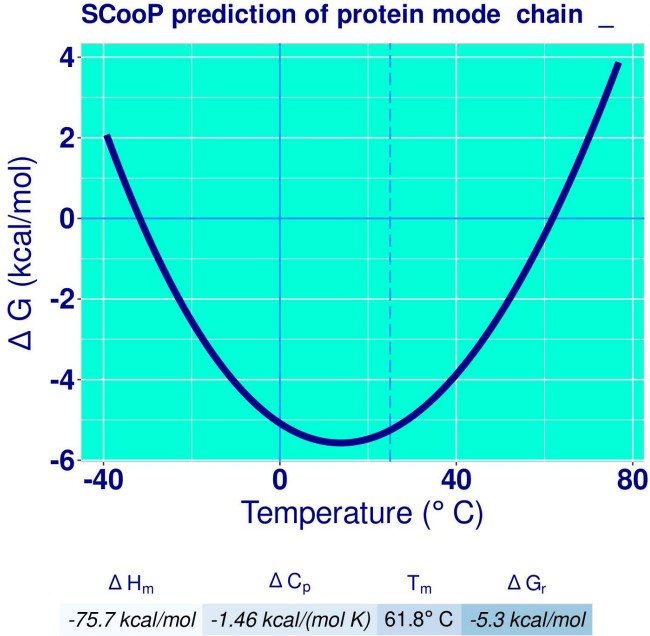

**SCooP prediction of protein mode chain _**

| Δ H$_m$ | Δ C$_p$ | T$_m$ | Δ G$_r$ |
|---|---|---|---|
| -75.7 kcal/mol | -1.46 kcal/(mol K) | 61.8° C | -5.3 kcal/mol |

**Fig 4. The stability of the final 3D structure of the vaccine, analyzed by SCooP.** The graph's vertical axis corresponds to the changes of Gibbs free energy, and the horizontal axis shows the temperature changes in °C. This graph shows the vaccine is stable at temperatures −30 to +60 °C.

**Table 6. Physicochemical properties of the final vaccine construct.**

| Characteristics | Value |
|---|---|
| Number of amino acids | 285 |
| Molecular weight | 31550.26 Da |
| Formula | $C_{1425}H_{2234}N_{390}O_{404}S_8$ |
| Total number of atoms | 4461 |
| Estimated half-life | 4.4 hours (mammalian reticulocytes, *in vitro*). <br>>20 hours (yeast, *in vivo*). <br>>10 hours (*Escherichia coli*, *in vivo*). |
| Aliphatic index | 93.33 |
| Grand average of hydropathicity (GRAVY) | −0.156 |
| Theoretical pI | 9.39 |
| Total number of negatively charged residues (Asp + Glu) | 18 |
| Total number of positively charged residues (Arg + Lys) | 30 |

After scrutinizing these thirty models with the PyMOL visualizer, the docked complexes with a proper interaction between the adjuvants and MD2 were selected and sorted according to their binding energy score. The chosen complexes were examined for the number and type of interactions between the amino acids of the adjuvants and MD2 obtained from the PDBsum (Fig 5B) and PPCHECK servers (Table 9). The complex with the best interactions, the lowest binding energy score, and the best dissociation constant (K$_D$) and binding free energy (ΔG), using PRODIGY analyses, was selected for further studies (Fig 5). The ΔG and KD according to the PRODIGY results were −16.6 and 1.8e-12, respectively.

**Table 7. The candidate residue pairs for the disulfide bond obtained from disulfide engineering.**

| Residue 1 | | | Residue 2 | | | Bond | | |
|---|---|---|---|---|---|---|---|---|
| Chain | Seq # | AA | Chain | Seq # | AA | $\chi_3$ | kcal/mol | Σ B-factor |
| A | 21 | LEU | A | 73 | ALA | +110.25 | 2.44 | 102.39 |
| A | 23 | ALA | A | 194 | PRO | +82.89 | 3.00 | 94.78 |
| A | 33 | LEU | A | 116 | ALA | +115.03 | 4.69 | 100.62 |
| A | 35 | LEU | A | 59 | ALA | −94.82 | 1.21 | 107.05 |
| A | 39 | SER | A | 59 | ALA | −92.00 | 2.55 | 107.35 |
| A | 42 | GLN | A | 55 | SER | +110.12 | 1.20 | 114.55 |
| A | 48 | CYS | A | 51 | LEU | +112.75 | 6.62 | 121.57 |
| A | 63 | LEU | A | 125 | LEU | +96.68 | 0.35 | 93.78 |
| A | 64 | TYR | A | 167 | ILE | +89.23 | 0.52 | 92.22 |
| A | 67 | PRO | A | 124 | LEU | −115.27 | 9.32 | 94.70 |
| A | 67 | PRO | A | 126 | PHE | +122.93 | 4.35 | 96.36 |
| A | 74 | ALA | A | 86 | LEU | +120.79 | 1.63 | 103.30 |
| A | 74 | ALA | A | 87 | LEU | +89.28 | 3.79 | 104.08 |
| A | 78 | MET | A | 83 | GLY | −82.93 | 3.03 | 96.02 |
| A | 98 | THR | A | 127 | TRP | +120.76 | 4.01 | 106.85 |
| A | 133 | CYS | A | 163 | ARG | +119.00 | 2.89 | 95.77 |
| A | 170 | ASP | A | 173 | TYR | +119.85 | 1.90 | 95.41 |
| A | 181 | LEU | A | 195 | PHE | +123.17 | 4.19 | 90.82 |
| A | 211 | HIS | A | 214 | LYS | +118.61 | 4.25 | 91.51 |
| A | 215 | ALA | A | 232 | SER | +90.91 | 5.07 | 114.50 |
| A | 226 | SER | A | 256 | SER | +120.21 | 7.14 | 151.36 |
| A | 229 | ALA | A | 253 | ALA | −101.60 | 3.84 | 146.57 |
| A | 248 | THR | A | 254 | TYR | −96.39 | 3.94 | 143.99 |

**Table 8. The conformational B-cell epitopes identified on the final vaccine 3D structure.**

| No. | Residues | No. of residues | Score |
|---|---|---|---|
| 1 | A:A1, A:P2, A:P3, A:H4, A:A5, A:L6, A:S7, A:E8, A:A9, A:A10, A:A11, A:K12, A:A13, A:K14, A:F15, A:V16, A:W19, A:M78, A:S79, A:L80, A:A81, A:S82, A:G83, A:K84, A:S85, A:H88 | 26 | 0.821 |
| 2 | A:A37, A:A40, A:D41, A:Q42, A:A43, A:L44, A:Q45, A:P46, A:P47, A:C48, A:P49, A:N50, A:L51, A:V52, A:S53 | 15 | 0.792 |
| 3 | A:I216, A:A218, A:T219, A:Q220, A:K221, A:K223, A:E224, A:T225, A:S226, A:S227, A:E228, A:A229, A:I230, A:S231, A:L234, A:K242, A:T248, A:D249, A:N250, A:G251, A:P252, A:A253, A:Y254, A:I255, A:S256, A:Q257, A:D258, A:F259, A:L260, A:N261, A:M262, A:C263, A:T264, A:S265, A:L266, A:A267, A:I268, A:H270, A:T271, A:T272, A:H273, A:E274, A:A275, A:A276, A:A277, A:K278, A:Q279, A:E280, A:I281, A:N282, A:S283, A:S284, A:Y285 | 53 | 0.754 |
| 4 | A:Q187, A:G188, A:R189 | 3 | 0.691 |
| 5 | A:S95, A:Q96, A:L97, A:T98, A:Q99, A:A100, A:I101, A:V102, A:K103, A:N104, A:H105, A:K106, A:N107, A:K110, A:I111, A:Y114, A:W127 | 17 | 0.69 |
| 6 | A:Q129, A:G130, A:G131, A:K134, A:E138, A:C141, A:F142 | 7 | 0.613 |
| 7 | A:G157, A:P158, A:G159, A:P160, A:G161, A:W162, A:R163 | 7 | 0.582 |
| 8 | A:N144, A:T146, A:N147, A:S151, A:Q154, A:E155 | 6 | 0.526 |
| 9 | A:E89, A:K92, A:R120, A:G121 | 4 | 0.514 |

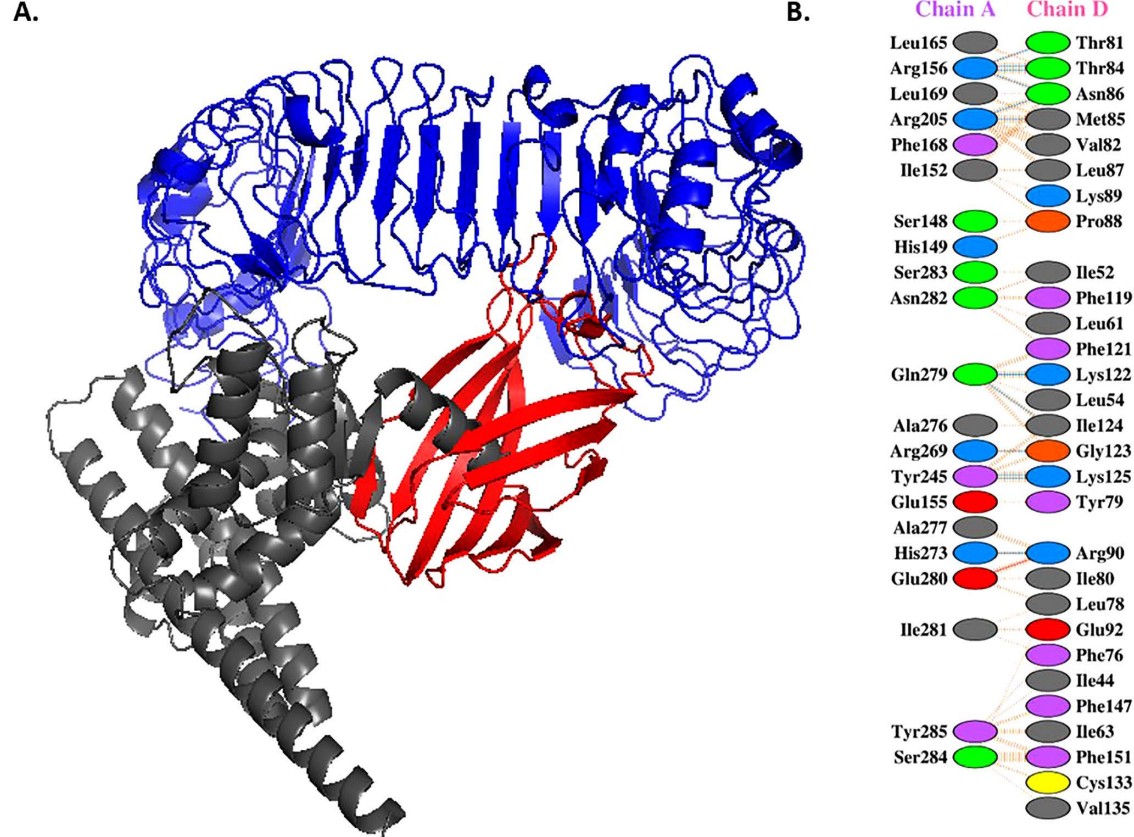

**Fig 5. Vaccine-TLR4-MD2 complex.** (A). The 3D structure of the selected complex of vaccine and TLR4/MD2, predicted by the ClusPro server. The gray, blue, and red structures are the protein vaccine, TLR4, and the MD2 co-receptor, respectively. (B) Amino acid binding map for the complex of MD2 co-receptor and vaccine obtained from the PDBsum server. guide: ▬▬ hydrogen bond, ▬▬ salt bridge, ||||||| non-bonded contacts between the vaccine and MD2 in the selected complex.

**Table 9. The type and number of bonds and interactions, suggested by PDBsum and PPCHECK.**

| Type of interactions | No. of interactions |
| --- | --- |
| Salt bridges | 1 |
| Hydrogen bonds | 12 |
| Non-bonded contacts | 151 |
| Hydrophobic interactions | 10 |
| Van der Waals Pairs | 3105 |
| Potential favorable electrostatic interactions | 3 |
| Potential unfavorable electrostatic interactions | 7 |

The ClusPro server also provided a value of 54 for the selected docked structure's members and a weighted score for the center and lowest energy of −985.8 and −1149.6, respectively.

**Molecular dynamics simulation analysis**

MD studies were conducted to investigate the behaviour of the vaccine-TLR4-MD2 complex during the simulation time. Fig 6 represents the RMSD of the three protein chains forming the complex structure, including vaccine, TLR4, and MD2, indicating

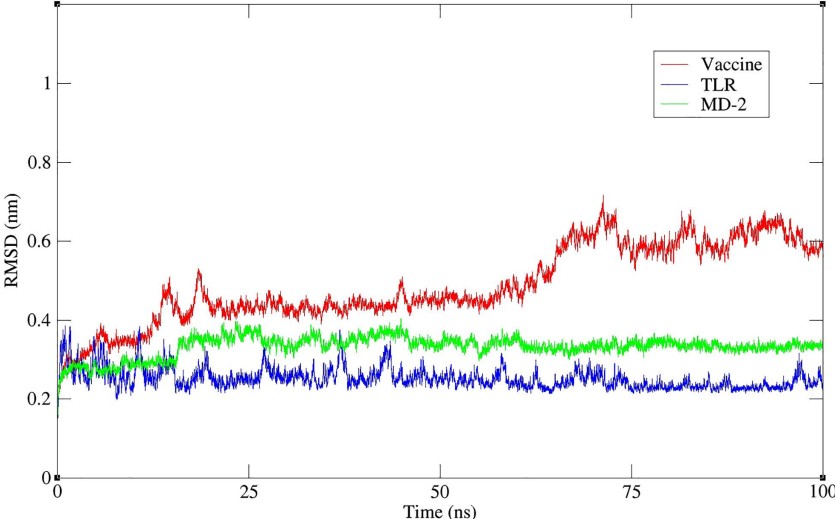

**Fig 6. Root Mean Square Deviation (RMSD) profiles of the docked vaccine construct (red), TLR4 (blue), and MD-2 co-receptor (green) throughout 100 ns molecular dynamics simulation.** RMSD values were calculated concerning the initial minimized structure for each component to evaluate their structural stability during the simulation. RMSD: Root Mean Square Deviation; TLR4: Toll-like receptor 4.

the stability of the structure's backbone atoms. The RMSD parameter presents the mean displacement of a protein's backbone atoms [33,34]. The RMSD graph of the MD simulation shows that vaccine proteins, TLR4, and MD2 had acceptable stability due to intramolecular and intermolecular interactions. Notably, during the simulation time, structural changes occur in these chains (especially for the vaccine) to find a stable position relative to the other two chains. These changes were bolder up to about 70 ns, after which the structures appeared to reach a stable state for the remainder of the simulation.

Fig 7 indicates the RMSF for each amino acid residue of the vaccine, TLR4, and MD2 molecules. This parameter measures the average deviation of atomic positions from their mean positions during the simulation and shows the residue's stability during the simulation analysis [33,34].

RMSF evaluation showed some fluctuations in all three structures and, to a greater extent, in the residues at each structure's N- and C-terminals. This is a typical indicator of the higher freedom of residues at the head and end of the molecules. TLR4 amino acids fluctuated less due to strong intramolecular and intermolecular interactions with MD2 and the vaccine. The highest peak in the RMSF is between amino acids 150 and 160, indicating the highest flexibility in this part of the molecule. This is due to the use of a flexible GPGPG linker in the range of amino acids 157–161.

The $R_g$, which indicates the level of compaction in the structure or how much the polypeptide chain is folded or unfolded, was also evaluated. It is obtained by calculating the mean square distance of atoms from the molecule's center of mass [62]. Fig 8 presents the $R_g$ plots of the three chains forming the complex, including vaccine, TLR4, and MD2. The $R_g$ values of the TLR4 protein decreased somewhat by the end of the simulation time, showing the molecule compression caused by strong intramolecular interactions. MD2 protein and vaccine had almost constant $R_g$ values during MD simulation time, which can represent their intermolecular solid interactions, strong intramolecular interactions in each chain, as well as the presence of intermolecular interactions in the complex, causing the molecules to be compressed, which results in $R_g$ reduction [33].

Hydrogen bonds formed between the vaccine and the TLR4 receptor were also investigated. Fig 9A and 9B show the number of residues involved in hydrogen bonds and the number of hydrogen bonds during the MD simulation, respectively. As shown in Fig 9B, the number of hydrogen bonds increased at the end of the simulation time compared to the beginning, indicating that the molecules had changed to reach a more stable position.

Fig 10 presents the interactions between TLR4 and MD2 with the vaccine's structure at its most stable state during the simulation time.

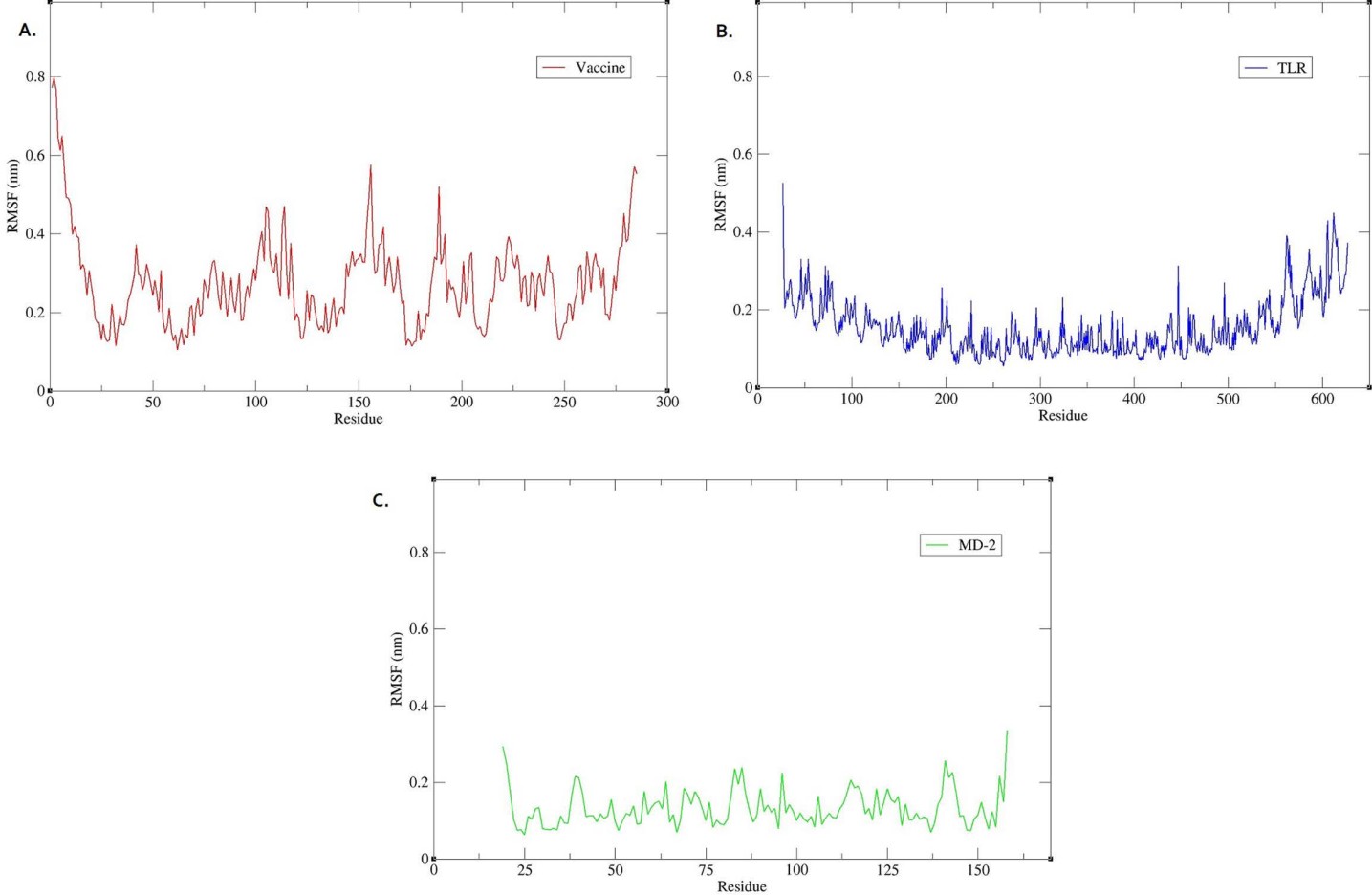

**Fig 7. The RMSF analysis of amino acid residues in the vaccine–TLR4–MD2 complex over the 100 ns molecular dynamics simulation.** (A) RMSF profile of the vaccine construct, highlighting regions with higher flexibility, likely corresponding to surface-exposed or loop regions. (B) RMSF profile of TLR4. (C) The RMSF profile of MD-2. RMSF: Root-mean-square fluctuation.

Fig 11 demonstrates the superimposition of the vaccine-TLR4-MD2 complex's 3D structure at the beginning and end of the simulation, showing some changes toward more stability.

## Population coverage prediction

The population coverage analysis of the selected overlapping epitopes showed a worldwide vaccine effectiveness of 77.44%. The highest coverage was found in Europe and North America (88.37% and 79.11%, respectively), and the lowest was in Central America (4.96%) (Table 10).

### *In silico* immune simulation

The C-ImmSim server simulated the vaccine's immune response by administering three doses (one injection every 28 days) in the human body. The results showed an increase in the level of IgM, which is the initial response. At the same time, as the concentration of antigens decreased, an increase in the level of B-cells and the secretion of IgM, IgG + IgM, and IgG1 + IgG2 antibodies was observed, especially after the second and third injections, which indicates the creation of

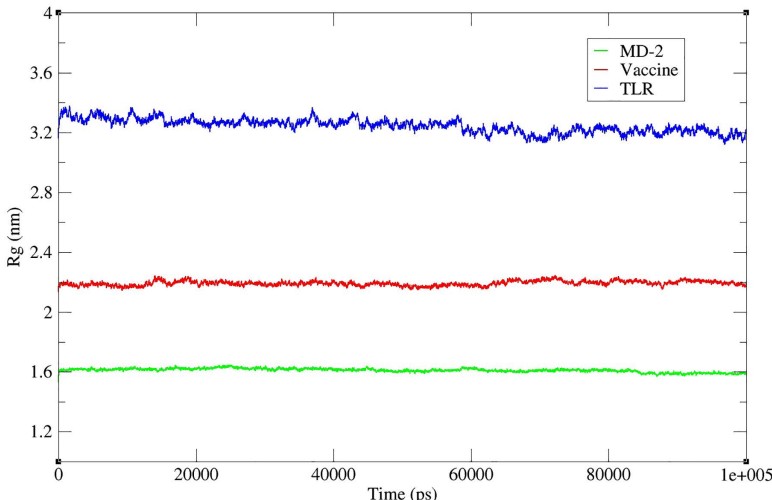

**Fig 8. Radius of gyration (Rg) profiles of the vaccine construct (red), TLR4 receptor (blue), and MD-2 co-receptor (green) over a 100 ns molecular dynamics simulation.** The Rg values indicate the overall compactness of each molecule during the simulation. The vaccine construct maintained an average Rg of approximately 2.2 nm, MD-2 around 1.6 nm, and TLR4 approximately 3.1 nm. The relatively stable Rg values for all three components suggest no major unfolding events and reflect structural compactness and stability throughout the simulation period.

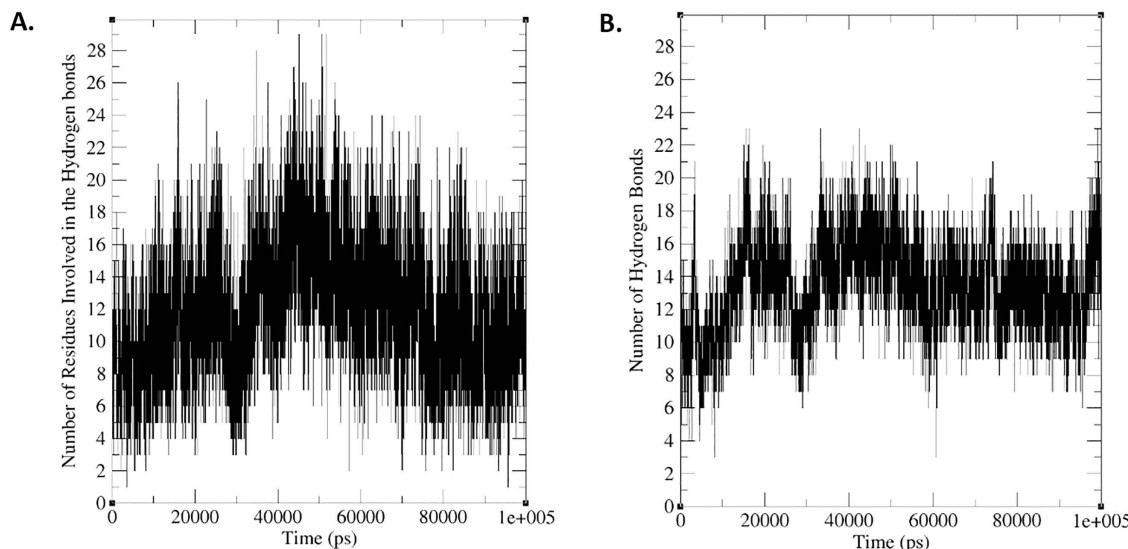

**Fig 9. Analysis of hydrogen bonding interactions within the vaccine–TLR4–MD-2 complex throughout the 100 ns molecular dynamics simulation.** (A) The number of residues participating in hydrogen bonding over the simulation time, indicating the extent of residue-level interactions contributing to complex stability. (B) Total number of hydrogen bonds formed during the simulation, reflecting the dynamic nature of intermolecular interactions.

immunological memory for future exposures with this virus. Moreover, the results proposed the higher activity of B- and T-cells and the persistence of memory B-cells for several months. The levels of cytokine secretion, especially IL-2 and IFN-γ, which play a key role in immune activation in facing viral infections, were also increased. Furthermore, macrophage and dendritic cell activities were adequate (Fig 12).

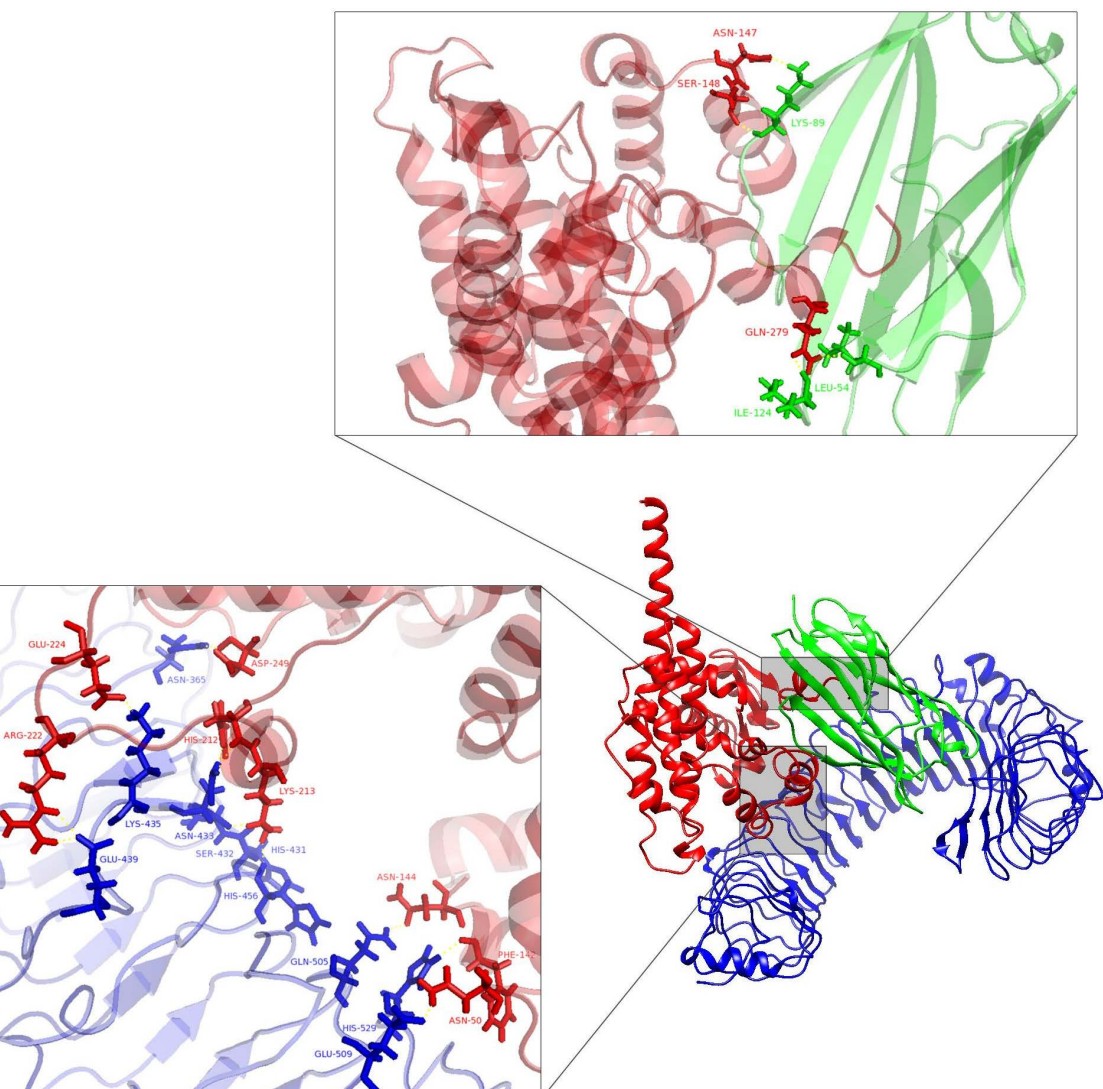

**Fig 10. The interactions between the 3D structure of the vaccine (red) with TLR4 (blue) and MD2 (green).** The structure is visualized by PyMOL 3.1.1. Vaccine residues N147, S148, and Q279 interact with MD2 residues K89, I124, and L54. TLR4 residues N365, L435, E439, N433, S432, H431, H456, Q505, H529, and E509 interact with vaccine residues R222, E224, D245, H212, L213, N144, F142, and N50.

## Codon optimization

The protein-coding sequence (CDS) of DNA codons was optimized using the GenSmart codon optimization tool to better translate the mRNA vaccine in the host's cells. CDS is a corresponding region of nucleotides with the amino acid sequence of a protein. After transcribing the optimized DNA sequence to RNA using the DNA>RNA>Protein tool, a CDS RNA sequence with 855 nucleotides was obtained. The rare codon analysis tool showed acceptable values for our mRNA vaccine. This vaccine with CAI = 0.73 is predicted to have high expression, given that CAI ≥ 0.7 is assumed to indicate high expression levels [63]. The optimum GC content of the sequence for sufficient expression is 30%−70%. Our construct's GC content was 54%, suggesting the vaccine's proper expression in human cells.

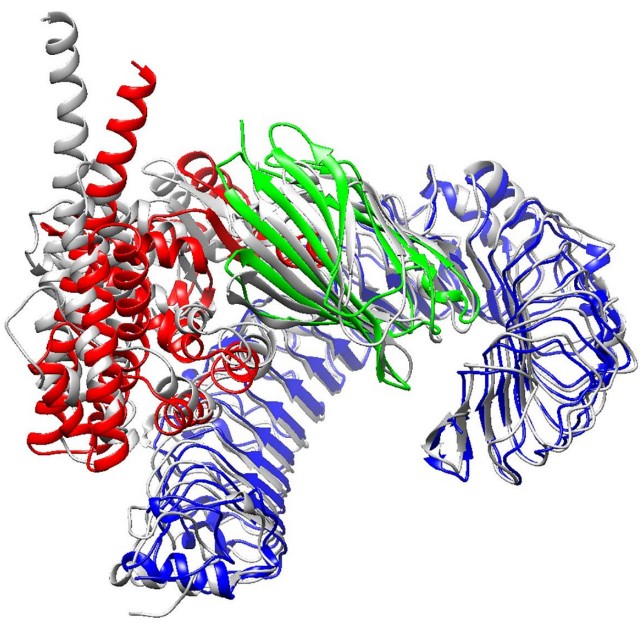

**Fig 11. Superimposition of the 3D structures of the complex at the beginning (structure with red, green, and blue colors) and at the end (gray structure) of the molecular dynamic simulation time.** The structures are visualized using PyMOL 3.1.1 and Chimera 1.17.3 MatchMaker module.

**Table 10. The worldwide population coverage of the designed vaccine.**

| Area | Population coverage |
| --- | --- |
| World | 77.44% |
| North America | 79.11% |
| Central America | 4.96% |
| South America | 42.20% |
| West Indies | 64.93% |
| Europe | 88.37% |
| North Africa | 59.80% |
| West Africa | 44.32% |
| Central Africa | 39.03% |
| South Africa | 44.17% |
| East Africa | 47.40% |
| Southwest Asia | 54.23% |
| South Asia | 51.90% |
| Southeast Asia | 25.54% |
| Oceania | 30.62% |
| East Asia | 38.80% |
| Northeast Asia | 28.89% |

The codon frequency distribution or CFD of our construct was 0.08. This factor indicates the percentage of rare codons (defined as <30% usage frequency), and several rare codons next to each other can decrease translation efficacy or even disengage the translational system [64].

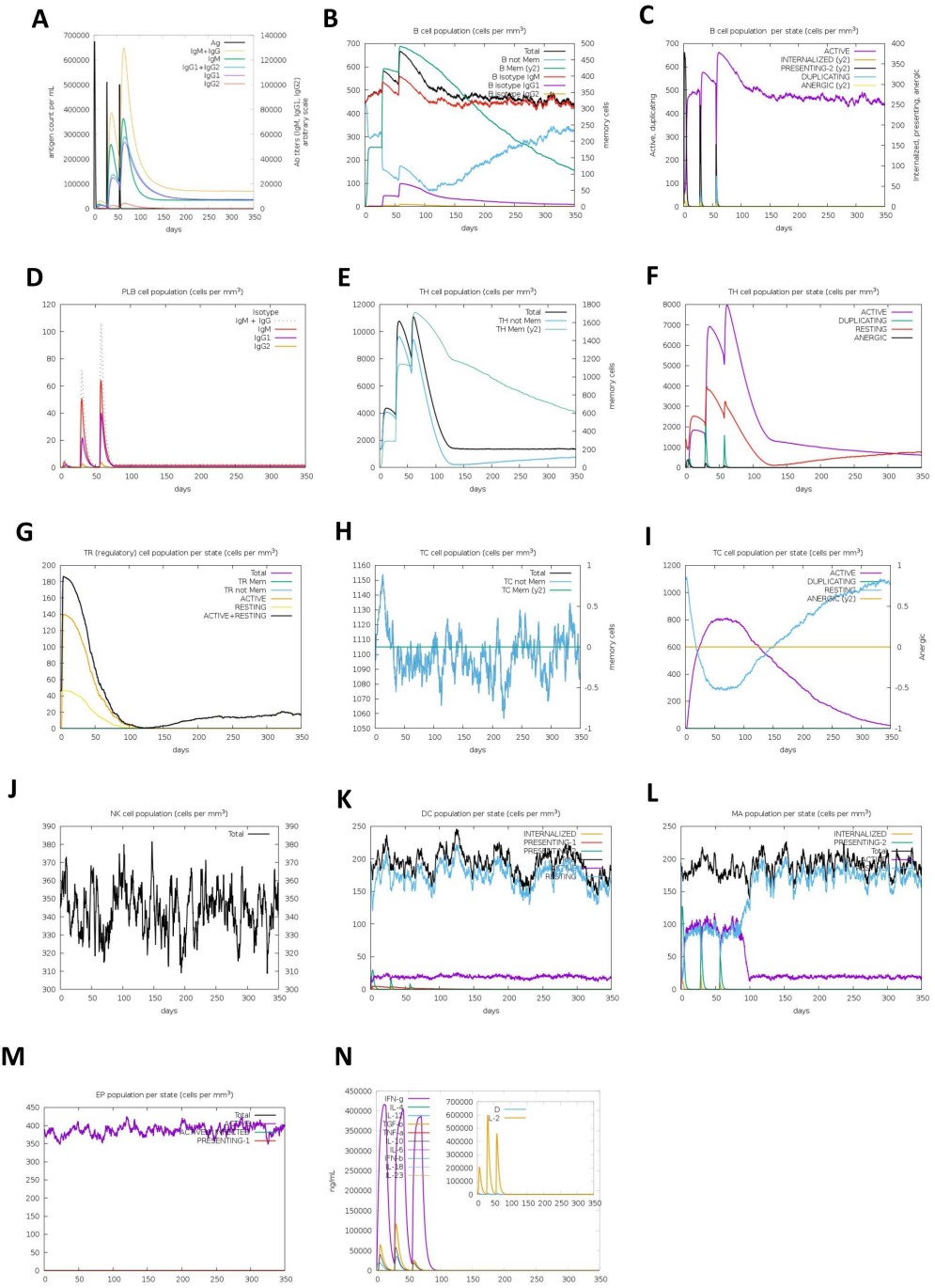

**Fig 12. Vaccine immune response simulated by the C-ImmSim server.** (A) Antigen and immunoglobulins. Antibodies are subdivided by isotype. (B) B lymphocytes: total count, memory cells, and subdivided isotypes (including IgM, IgG1, and IgG2). (C) B lymphocyte population per entity-state (i.e., showing counts for active, presenting on class-II, internalized the antigen, duplicating, and anergic. (D) Plasma B lymphocytes count subdivided per isotype (IgM, IgG1, and IgG2). (E) CD4 T-helper lymphocytes count. The plot shows total and memory counts. (F) CD4 T-helper lymphocytes count subdivided per entity-state (i.e., active, resting, anergic, and duplicating). (G) CD4 T-regulatory lymphocytes count. Total, memory, and per-entity-state counts are plotted here. (H) CD8 T-cytotoxic lymphocytes count (Total and memory). (I) CD8 T-cytotoxic lymphocytes count per entity-state. (J) Natural Killer cells (total count). (K) DCs can present antigenic peptides on both MHC class I and class II molecules. The curves show the total number broken down into active, resting, internalized, and presenting antigens. (L) Macrophages. Total count, internalized, presenting on MHC class-II, active and

resting macrophages. (M) Epithelial cells. The total count is broken down to active, virus-infected, and presenting on class-I MHC molecules. N. Cytokines. Concentration of cytokines and interleukins. D in the inset plot is a danger signal. DC: Dendritic cell.

## mRNA construct

Fig 13 schematically demonstrates the final mRNA construct segments from the N-terminal to the C-terminal.

## RNA vaccine secondary structure

The secondary structures of the mRNA and their minimum free energy (MFE) were predicted by RNAfold. The MFE and centroid secondary structure showed a minimum free energy of −2049.70 kcal/mol and −1793.84 kcal/mol, respectively (Fig 14), indicating the mRNA could stay stable after production.

## PTM analyses

Several PTM analyses were performed on the final peptide vaccine. Evaluating the results of NetNGlyc-1.0, NetPhos-3.1, and NetAcet-1.0 servers showed one N-glycosylation site, 32 phosphorylation sites (serine: 18, threonine: 9, tyrosine: 5), and no N-acetylation site in the construct. Additionally, no potential GPI modification site and no myristoylation site accessible for NMT were detected by the big-PI/GPI animal and MyrPS/NMT servers (Table 11).

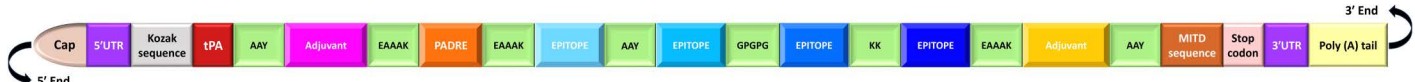

**Fig 13. Schematic representation of the designed mRNA vaccine sequence.**

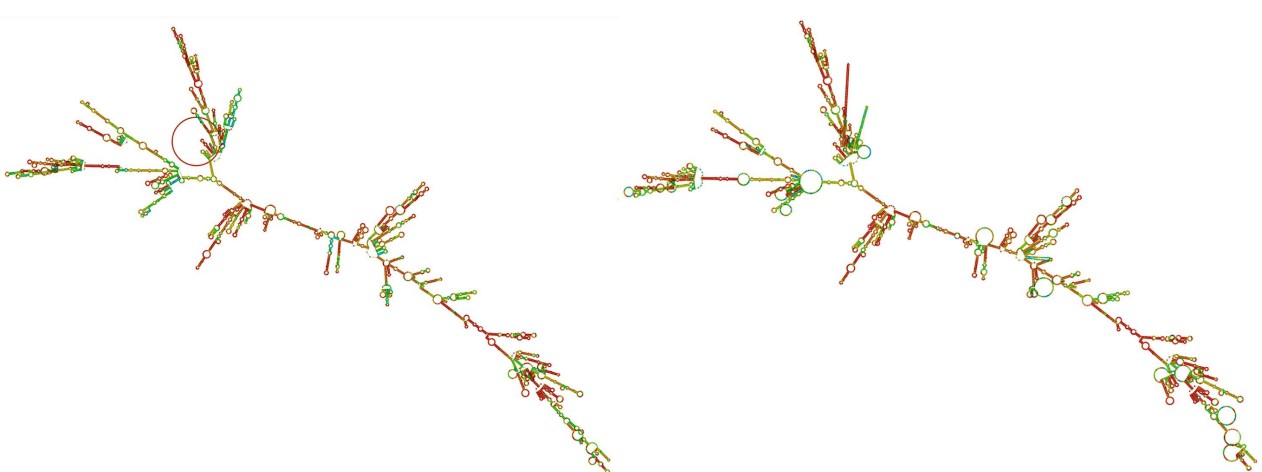

**Fig 14. MFE secondary structure (left) and centroid secondary structure (right) of the mRNA vaccine in positional entropy format, shown in red-colored bases (low entropy, defined) to green to blue and purple (high entropy, uncertain) bases.** For example, red areas are correctly predicted, but blue or green areas are not completely correct and regular.

**Table 11. Post-translational modification of the final vaccine via different servers.**

**Specific post-translational modification**

| | Prediction of a specific function | Server | Result |
|---|---|---|---|
| Lipid post-translational modifications (PTMs) | GPI modification site prediction | Big-PI/GPI animals | No potential GPI modification site |
| | Myristoyl | MyrPS/NMT | No myristoylation site |
| Phosphorylation | Identification of the general phosphorylation site | NetPhos-3.0 | 32 phosphorylation sites (Serine: 18, Threonine: 9, Tyrosine: 5) |
| Glycosylation | N-glycosylation sites | NetNGlyc-1.0 | One N-glycosylation site |
| Acetylation | N-terminal acetylation | NetAcet-1.0 | No significant N-terminal acetylation sites |

## Discussion

HTLV-1 can cause several diseases with poor prognosis, while efficacious therapeutic choices are limited [65]. Vaccination is an effective and safe strategy to protect against infectious diseases for years, decades, or even a lifetime [17]. In this study, an mRNA vaccine against HTLV-1 was designed *in silico* using "reverse vaccinology" approaches [66] by multiple computational and ML- and AI-based algorithms.

While both prophylactic and therapeutic vaccines against HTLV-1 are needed, here we aimed to design a prophylactic mRNA vaccine. Theoretically, our candidate vaccine could be used for both purposes based on the employed key epitopes. The performed immune simulation of our designed vaccine candidate also showed the induction of both humoral and cellular immunity responses needed to combat the virus in various disease phases. However, due to the complexity and limited knowledge on therapeutic effectiveness, we mainly focused on this construct as a prophylactic vaccine candidate.

HTLV-1 is a single-stranded RNA retrovirus. Due to undergoing a reverse-transcription step, its RNA genome can be converted to DNA and integrated into the host chromosome. HTLV-1 persistence mainly occurs through infected cell proliferation, so preventing initial infection is crucial. Once infection is established, even potent neutralizing antibodies may have limited effect. Thus, most vaccines designed against this virus are intended for prevention [12]. However, in 2015, a study evaluated the safety and effectiveness of a Tax peptide-pulsed dendritic cell therapeutic vaccine on three patients with ATL. All three patients showed Tax-specific CTL responses that peaked at 16–20 weeks after vaccination [12,67]. The employed vaccine platform might justify its therapeutic targeted indication. Our vaccine contains the Env viral proteins, which have been previously used for both preventive (as chimeric multiepitope based vaccine) [68] and therapeutic (as epitope-based vaccine) [69] purposes.

The selection of mRNA platform in our study was based on the general efficacy and balanced strong immune responses observed for these vaccines. Noteworthy, synthetic mRNA used in vaccines remains confined to the cytoplasm, does not undergo reverse transcription, and lacks any mechanism for genomic integration. This transient expression of viral antigens is rapidly degraded after translation, while still capable of inducing effective immune responses. Therefore, mRNA vaccination does not pose risks of contributing to HTLV-1–like integration events and remains a theoretically viable approach against HTLV-1. Additionally, HTLV-1 integrase, the viral enzyme responsible for proviral integration into the host DNA, limits the effectiveness of vaccine-induced immunity, especially humoral responses. Thus, strong cellular immune responses are required. Integrase inhibitors (originally developed for HIV-1) have shown potential to disrupt this process in HTLV-1, suggesting that their therapeutic use could complement vaccination strategies by limiting reservoir formation and enhancing sustained viral control [70].

Several studies have designed peptide-based vaccines against HTLV-1. Raza et al. designed an *in silico* HTLV-1 polypeptide vaccine, utilizing the Tax viral protein [69]. Another peptide vaccine was designed using viral non-structural (HBZ and Tax) and viral envelope (gp46) proteins [12]. A previous study also employed the gp46 peptide sequence (part of the gp62 sequence used in our study) and biodegradable microspheres (to eliminate booster immunization) to design a

peptide vaccine against HTLV-1. This vaccine elicited high antibody titers in rabbits and mice [71]. However, the main limitation of peptide-based vaccines is their lower immunogenicity [34]. In our study, we employed gp64 to design an mRNA vaccine, which could potentially be more efficient.

Vaccination success depends on the appropriate use of specific antigens, called immunodominant epitopes. Thus, it is vital to predict immunodominant epitopes that can stimulate both B- and T-cells [17,72]. Here, we first evaluated the antigenicity of eight viral proteins. These eight proteins, with oncogenic, structural, or regulatory roles, are crucial for viral replication and pathogenicity [10]. Then, the two of these antigenic proteins with higher antigenicity were selected for epitope prediction. With a precise evaluation of the predicted epitopes' overlaps, three and two overlapped regions were chosen from gp62 and pol, respectively. These five candidate regions included high-scoring epitopes from the four evaluated epitope types. Four out of five segments were finally selected by assessing allergenicity, antigenicity, and toxicity [73].

To enhance immune responses, in addition to selecting the most antigenic epitopes, three adjuvants were employed in our designed constructs. The used adjuvants were pan HLA DR-binding epitope (PADRE) and three TLR4 agonists: mycobacterial heparin-binding hemagglutinin (HBHA) conserved [32], RS01, and RS09 (lipopolysaccharide mimotope synthetic peptides) [35].

TLR4 agonists such as beta-defensin (hBds), PADRE [74], and RS09 [75] have been used in similar vaccine-design studies against this virus. PADRE, a synthetic peptide with 13 amino acids, is a more potent universal T helper epitope than other known T helper cell epitopes. It has several benefits, including safety for humankind in the clinic and the ability to induce CD4+Th cells effectively and bind to most HLA-DR molecules [76,77].

In a similar study, an mRNA vaccine against the HTLV-1 virus was designed, and its immunogenicity was evaluated in New Zealand rabbits. They used a delivery system called lipid nanoparticle (LNP), which also has immunogenic properties and can function as an adjuvant [65,78]. Here, we employed built-in adjuvants; though, delivery systems might be needed in the future formulation of our vaccine.

Another crucial step in designing a multi-epitope vaccine is choosing an appropriate linker, as it can impact the properties of the resulting protein and optimize antigen presentation. A lack of linkers may result in neoepitopes or junctional epitopes, creating a new protein with unknown properties [34]. In vaccine design, it is necessary to ensure that the domains work independently without interfering with each other to induce the targeted immune system receptors. Generally, linkers are selected based on their rigidity/flexibility and length factors [17,76,79]. GPGPG [30], KK [31], and AAY [32] peptide linkers were used here to join epitopes. The adjuvants were linked to the other parts by the EAAAK rigid linker [76]. Benefits of EAAAK include achieving high expression and improving bioactivity and stability [76].

Docking results and MD simulation analyses showed suitable engagement of the vaccine with TLR4/MD2. Evaluation of the complex 3D structure shows the involvement of RS01, the TLR4 agonist adjuvant, in interaction with the MD2 co-receptor, which is strengthened during the simulation. In addition, the MD2 residues involved in the interaction were similar to the pattern of the natural interaction of TLR4-MD2 with LPS [34].

Following codon optimization and transcription to RNA, several segments, including 3'- and 5'-UTR (untranslated region), 5'-cap, MITD sequence, poly(A) tail, and a tissue plasminogen activator (tPA) signal peptide, were added to design the final mRNA vaccine construct. A brief rationale for each added element is given below.

The cap at the 5′-terminal of the mRNA strand is necessary for being identified by eukaryotic translation initiation factor 4E, protection from degradation through exonuclease attack, non-self versus self-recognition, and triggering the translation initiation complex of ribosomes. Different natural types of 5′ cap structures have already been discovered [80]. We added the 5′-cap using the ARCA (7-methyl(3-O-methyl) GpppG Cap).

Generally, the main role of mRNA UTRs is post-transcriptional control of gene expression. Here, the human β-globin gene and the rabbit β-globin gene were used as 5′-UTR and 3′-UTR, respectively [17,80].

The open reading frame (ORF) of mRNA includes translatable sequences of the desired protein and higher-order RNA elements that affect translation efficacy. The mRNA vaccine construct should contain five elements in its ORF, including

the Kozak sequence, adjuvant, overlapping epitopes, linkers, and stop codon [17,80]. To enhance the immunogenicity of the mentioned mRNA vaccine, the tissue plasminogen activator (tPA), a secretory signal peptide of *Homo sapiens* [81], was connected to the 5′ region of the ORF using an AAY linker. To amplify the immune response or, more specifically, to enhance the efficiency of CD4 + T-cell antigen presentation, an MHC I-targeting domain (MITD) was added to the 3′ region of the ORF using an AAY linker [81].

The poly(A) tail typically contains 60–150 nucleotides, and the stability (half-life) and translation efficiency of mRNA are affected by its length [80,82]. A poly(A) tail was used, containing 120 nucleotides (A120), to support protein expression and lengthen the mRNA vaccine. The Pfizer/BioNTech mRNA vaccine contains two segmented poly(A) tails with a small spacer [83].

The PTM analysis demonstrated 32 phosphorylation sites and only one glycosylation modification site. Several advantages of phosphorylation sites include better degradation of mRNA, entry into the MHC-I pathway, high affinity for cytotoxic T lymphocytes, and boosting immune responses [84]. Thus, the presence of 32 phosphorylation sites in our vaccine candidate might potentially serve its functionality.

Currently, no vaccine is licensed for HTLV-1, and no candidate has demonstrated proven efficacy and safety in humans [3]. Given the long-term intracellular persistence and the virus's ability to evade classical antiviral immunity, vaccine development has been challenging. While mRNA vaccine approaches are under investigation, claims of prophylactic protection against HTLV-1 require robust clinical evidence beyond *in silico* predictions or preclinical models. Given this gap, the strongest public health measure presently remains the prevention of HTLV-1 transmission through established strategies in blood safety, maternal and neonatal health, sexual health education, and risk-reduction practices. A dual strategy, namely, accelerating rigorous vaccine development while strengthening transmission-prevention measures, offers the most practical path toward HTLV-1/2 control and eventual elimination in future [85].

## Limitations

This study presents a computational design for an mRNA-based vaccine targeting HTLV-1 as a suggested preventive solution in endemic regions. However, several limitations must be acknowledged. The findings are based solely on *in silico* analyses, and experimental validation is required to confirm antigen stability, delivery efficiency, and immunogenic potential. These limitations highlight the need for further preclinical and clinical investigations to assess the feasibility of this approach.

## Conclusion

An mRNA vaccine candidate containing a TLR4 agonist adjuvant and PADRE epitope was designed against HTLV-1 using AI- and ML-driven reverse vaccinology approaches. Different performed analyses predicted its potential immunogenicity, physicochemical, and immunogenic characteristics after translation into protein in the body, as well as solid interactions between the vaccine and TLR4/MD2. However, further *in vitro* and *in vivo* experimental research is needed to confirm the efficacy of our mRNA vaccine candidate against HTLV-1.

## Supporting information

**S1 Fig. Position of discontinuous conformational B cell epitopes (yellow regions) on the 3D structure of the translated protein from the vaccine.**
(TIFF)

**S1 Table. The sequences of HTLV-1 proteins.**
(DOCX)

**S2 Table. Primary six suggested constructs of protein vaccine.**
(DOCX)

**S3 Table. The evaluation results of 3D structures for six proposed 3D models.**
(DOCX)

**S4 Table. Validation results of the refined 3D structures of the selected structure in the previous step.**
(DOCX)

**S5 Table. The impact of mutations of HIS 105 and VAL 207 on the vaccine's stability.**
(DOCX)

**S6 Table. URL and validity of the used servers in this study.**
(DOCX)

## Author contributions

**Conceptualization:** Navid Nezafat, Manica Negahdaripour.

**Formal analysis:** Nadia Seifi, Mohammad Soroosh Hajizade, Manica Negahdaripour.

**Funding acquisition:** Navid Nezafat.

**Investigation:** Nadia Seifi, Manica Negahdaripour.

**Methodology:** Navid Nezafat, Manica Negahdaripour.

**Software:** Mohammad Soroosh Hajizade, Manica Negahdaripour.

**Supervision:** Navid Nezafat, Manica Negahdaripour.

**Validation:** Nadia Seifi, Manica Negahdaripour.

**Visualization:** Nadia Seifi, Mohammad Soroosh Hajizade.

**Writing – original draft:** Nadia Seifi, Manica Negahdaripour.

**Writing – review & editing:** Navid Nezafat, Manica Negahdaripour.

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
