## [Decision Letter · Decision Letter 0]

4 Feb 2025

PONE-D-25-00727Design and in silico evaluation of an mRNA vaccine against HTLV-1 using AI-driven reverse vaccinology approachesPLOS ONE

Dear Dr. Negahdaripour,

Thank you for submitting your manuscript to PLOS ONE. After careful consideration, we feel that it has merit but does not fully meet PLOS ONE’s publication criteria as it currently stands. Therefore, we invite you to submit a revised version of the manuscript that addresses the points raised during the review process.

We look forward to receiving your revised manuscript.

Kind regards,

Sheikh Arslan Sehgal, PhD

Academic Editor

PLOS ONE

Journal Requirements:

2. Please note that PLOS ONE has specific guidelines on code sharing for submissions in which author-generated code underpins the findings in the manuscript. In these cases, we expect all author-generated code to be made available without restrictions upon publication of the work.

Please review our guidelines at https://journals.plos.org/plosone/s/materials-and-software-sharing#loc-sharing-code and ensure that your code is shared in a way that follows best practice and facilitates reproducibility and reuse.

“This work was supported by the Vice-Chancellor for Research, Shiraz University of Medical Sciences, Iran (Grant number: 28569).”

4. Please note that funding information should not appear in the Acknowledgments section or other areas of your manuscript. We will only publish funding information present in the Funding Statement section of the online submission form. Please remove any funding-related text from the manuscript.

5. We note that Figure 12 in your submission contain map images which may be copyrighted. All PLOS content is published under the Creative Commons Attribution License (CC BY 4.0), which means that the manuscript, images, and Supporting Information files will be freely available online, and any third party is permitted to access, download, copy, distribute, and use these materials in any way, even commercially, with proper attribution. For these reasons, we cannot publish previously copyrighted maps or satellite images created using proprietary data, such as Google software (Google Maps, Street View, and Earth). For more information, see our copyright guidelines: http://journals.plos.org/plosone/s/licenses-and-copyright.

1) You may seek permission from the original copyright holder of Figure 12 to publish the content specifically under the CC BY 4.0 license.

Please upload the completed Content Permission Form or other proof of granted permissions as an " "Other" " file with your submission.

2) If you are unable to obtain permission from the original copyright holder to publish these figures under the CC BY 4.0 license or if the copyright holder’s requirements are incompatible with the CC BY 4.0 license, please either i) remove the figure or ii) supply a replacement figure that complies with the CC BY 4.0 license. Please check copyright information on all replacement figures and update the figure caption with source information. If applicable, please specify in the figure caption text when a figure is similar but not identical to the original image and is therefore for illustrative purposes only.

Reviewers' comments:

Reviewer's Responses to Questions

**Comments to the Author**

1. Is the manuscript technically sound, and do the data support the conclusions?

Reviewer #1: Partly

Reviewer #2: Partly

2. Has the statistical analysis been performed appropriately and rigorously? 

Reviewer #1: Yes

Reviewer #2: N/A

3. Have the authors made all data underlying the findings in their manuscript fully available?

Reviewer #1: Yes

Reviewer #2: Yes

4. Is the manuscript presented in an intelligible fashion and written in standard English?

Reviewer #1: Yes

Reviewer #2: No

5. Review Comments to the Author

Reviewer #1: The study entitled “Design and in silico evaluation of an mRNA vaccine against HTLV-1 using AI-driven reverse vaccinology approaches” by Seifi et al., The study has used an artificial intelligence-driven reverse vaccinology approach to design a mRNA vaccine against HTLV-1. Therefore, this study can be recommended for publication if the following comments are addressed.

1. Explain the reason behind selecting the HTLV-1 target proteins as “candidate proteins” to design the vaccine construct.

2. What specific factors were considered to optimize antigen presentation and enhance immune responses, such as epitope selection, adjuvant usage, or linker design in the vaccine construct?

3. Disulfide bridges play a crucial role in stabilizing the protein’s folded conformation; therefore, the authors are encouraged to include a section on “Disulfide engineering of the vaccine”.

4. The authors are advised to assess the solubility of the proposed vaccine construct, as solubility is a critical factor influencing its stability and bioavailability.

5. Explain the reason in discussion why three different linkers are used to construct the vaccine.

6. The Ramachandran plot of the modelled vaccine structure shows one outlier residue (His105). Could you provide insights into its role in the vaccine structure, and whether it impacts the stability or functionality of the vaccine? Additionally, were any efforts made to refine or resolve this outlier during structure validation?

7. Perform in silico cloning for maximum vaccine expression and to predict the reverse translation efficacy.

8. The discussion section also contains several statements that require relevant explanation and citations.

9. The manuscript mentions the use of artificial intelligence (AI) to construct the vaccine, but the described methodology appears to rely on online servers and databases. Could the authors elaborate on the specific role AI played in the vaccine design process? If AI was not directly employed, consider revising the description for clarity.

10. The authors should use other studies conducted in this field in this section.

https://doi.org/10.3390/v14112504

https://doi.org/10.1007/s12026-023-09403-2

https://doi.org/10.1016/j.vacun.2024.04.003

https://doi.org/10.1007/s13205-024-04022-6

https://doi.org/10.1016/j.biologicals.2024.101782

Reviewer #2: 1. In the method section, authors must include the access link to the servers used in front of their names.

2. In this work, only epitopes related to a specific set of alleles are predicted, and the authors must provide a logical reason for this.

3. The authors should perform disulfide engineering of the vaccine construct.

4. The captions for the simulation results figures are very brief. Please provide more explanation about these figures.

5. Please revise the English language of your manuscript

6. PLOS authors have the option to publish the peer review history of their article (what does this mean? ). If published, this will include your full peer review and any attached files.

**Do you want your identity to be public for this peer review?** For information about this choice, including consent withdrawal, please see our Privacy Policy .

Reviewer #1: **Yes:** Dr. Anand Anbarasu

Reviewer #2: No

---

## [Author Response · Author response to Decision Letter 1]

21 May 2025

Dear Editor,

Your comments and the reviewer's insights were highly insightful and enabled us to greatly improve the quality of our manuscript. The following pages are our amendments on journal requirements and our one-by-one responses to each of the reviewers' comments.

Revisions in the text are shown using track changes and highlights. Furthermore, the clear version of the revised manuscript (without track changes) was uploaded to the e-Component section.

Journal Requirements:

Done accordingly.

2. Please note that PLOS ONE has specific guidelines on code sharing for submissions in which author-generated code underpins the findings in the manuscript. In these cases, we expect all author-generated code to be made available without restrictions upon publication of the work.

Please review our guidelines at https://journals.plos.org/plosone/s/materials-and-software-sharing#loc-sharing-code and ensure that your code is shared in a way that follows best practice and facilitates reproducibility and reuse.

Not applicable.

“This work was supported by the Vice-Chancellor for Research, Shiraz University of Medical Sciences, Iran (Grant number: 28569).”

Done accordingly.

4. Please note that funding information should not appear in the Acknowledgments section or other areas of your manuscript. We will only publish funding information present in the Funding Statement section of the online submission form. Please remove any funding-related text from the manuscript.

Done accordingly.

5. We note that Figure 12 in your submission contain map images which may be copyrighted. All PLOS content is published under the Creative Commons Attribution License (CC BY 4.0), which means that the manuscript, images, and Supporting Information files will be freely available online, and any third party is permitted to access, download, copy, distribute, and use these materials in any way, even commercially, with proper attribution. For these reasons, we cannot publish previously copyrighted maps or satellite images created using proprietary data, such as Google software (Google Maps, Street View, and Earth).

The figure was deleted, and the data is now presented in a table (Table 10).

Reviewer 1:

1. Explain the reason behind selecting the HTLV-1 target proteins as “candidate proteins” to design the vaccine construct.

✔ Thank you very much for your insightful comments. Eight candidate proteins were considered based on their previous application in vaccine design in earlier studies. The rationale for using these eight proteins in previous studies was that they possess oncogenic, structural, or regulatory functions and play essential roles in the viral replication cycle and pathogenicity (Page 36 in the revised manuscript with track change 34 in the clean manuscript). Then, based on antigenicity and allergenicity assessments, the Pol and GP62 antigens were selected for the final epitope selection.

2. What specific factors were considered to optimize antigen presentation and enhance immune responses, such as epitope selection, adjuvant usage, or linker design in the vaccine construct?

✔ Various techniques were used to improve antigen presentation and immune responses. For antigen presentation, the use of flexible linkers between epitopes was employed. To enhance immune responses, it was done by using the most antigenic epitopes from the most antigenic proteins in vaccine design, along with the application of various adjuvants such as RS01, RS09, PADRE, and conserved HBHA. Moreover, adjuvants were used as built-in components of the vaccine construct. Some more related explanations were added in the discussion to highlight these points (pages 35).

3. Disulfide bridges play a crucial role in stabilizing the protein’s folded conformation; therefore, the authors are encouraged to include a section on “Disulfide engineering of the vaccine”.

✔ We added Disulfide engineering of our vaccine, and this section is included in the manuscript (Methods: page 10).

4. The authors are advised to assess the solubility of the proposed vaccine construct, as solubility is a critical factor influencing its stability and bioavailability.

✔ Thanks for your suggestion. Since the vaccine is not protein and will not be produced by cloning in E. coli, the usual solubility servers like SolPro, which predicts protein solubility upon overexpression in E. coli, would not be suitable. We added solubility analysis by Aggrescan3D, which anticipates the aggregative properties of protein and fits our purpose better (Methods: page 11).

5. Explain the reason in discussion why three different linkers are used to construct the vaccine.

✔ Both rigid and flexible linkers were utilized, including GPGPG, KK, AYY, and EAAAK. Flexible linkers can be used between epitopes to reduce junctional immunogenicity. They also contribute to the immunogenicity of multi-epitope vaccines. However, a very high protein flexibility might harm the stability of the 3D structure, so we used a complex of several different linkers. For linking adjuvants to other components, a rigid linker (EAAAK) was employed, which helps increase expression, stabilize the structure, and improve bioactivity.

6. The Ramachandran plot of the modelled vaccine structure shows one outlier residue (His105). Could you provide insights into its role in the vaccine structure, and whether it impacts the stability or functionality of the vaccine? Additionally, were any efforts made to refine or resolve this outlier during structure validation?

✔ The impact of substituting the mentioned amino acid (His105), as well as another amino acid in a similar context at position 207 (Valine), with 19 other amino acids was evaluated using the CUPSAT server. The effects of these substitutions on protein structural stability were summarized in tables 7 & 8, based on the calculated ΔΔG values.

7. Perform in silico cloning for maximum vaccine expression and to predict the reverse translation efficacy.

Since the vaccine construct is mRNA, not protein, the classical in silico cloning might not apply here, but DNA codon optimization was done by GenSmart Codon Optimization, and factors such as CAI, CFD, and GC content were calculated (page 31).

8. The discussion section also contains several statements that require relevant explanation and citations.

✔ The discussion was reviewed precisely, and every sentence needing a reference was cited. Added citations are shown in yellow highlights.

9. The manuscript mentions the use of artificial intelligence (AI) to construct the vaccine, but the described methodology appears to rely on online servers and databases. Could the authors elaborate on the specific role AI played in the vaccine design process? If AI was not directly employed, consider revising the description for clarity.

✔ AI is categorized as predictive, descriptive, prescriptive, or generative. Machine learning, a subset of AI, develops algorithms that improve with experience. Deep learning, a form of ML, uses neural networks to identify complex patterns (Gasperini, Baylor et al. 2025). A number of servers used in this paper use artificial neural network (ANN) based or ML approaches to predict the mentioned results.

* Gasperini, G., et al. (2025). "Vaccinology in the artificial intelligence era." Science Translational Medicine 17(794): eadu3791.

We revised several related sentences as you suggested. (Discussion: page 34/ Introduction: page 5)

10. The authors should use other studies conducted in this field in this section.

https://doi.org/10.3390/v14112504

https://doi.org/10.1007/s12026-023-09403-2

https://doi.org/10.1016/j.vacun.2024.04.003

https://doi.org/10.1007/s13205-024-04022-6

https://doi.org/10.1016/j.biologicals.2024.101782

✔ The mentioned references were used to improve the manuscript (ref: 11, 67, 72, 73, 76).

Reviewer 2:

1. In the method section, authors must include the access link to the servers used in front of their names.

✔ Thank you very much for your helpful comments. The access links have been added accordingly.

2. In this work, only epitopes related to a specific set of alleles are predicted, and the authors must provide a logical reason for this.

✔ Prediction of HLA I and HLA II binding epitopes were done for three MHC class I supertypes, namely A*01:01, A*02:01, and A*03:01, and three MHC class II supertypes, including DRB1*01:01, DRB1*03:01, and DRB1*04:01. These epitopes were selected to optimize population coverage, taking into account their wide distribution and overlap across various populations (Methods: page 6).

3. The authors should perform disulfide engineering of the vaccine construct.

✔ We added Disulfide engineering of our vaccine, and this section is included in the manuscript (Methods: page 10).

4. The captions for the simulation results figures are very brief. Please provide more explanation about these figures.

✔ Captions were edited, and more explanations were added. Thank you for this helpful suggestion.

5. Please revise the English language of your manuscript

✔ The manuscript was carefully reviewed, and all grammatical and punctuation errors were corrected.

Again, we appreciate all your insightful comments. We worked hard to be responsive to them. Thank you for taking the time and energy to help us improve the paper.

Sincerely yours,

The corresponding author,

Manica Negahdaripour

---

## [Decision Letter · Decision Letter 1]

19 Jun 2025

PONE-D-25-00727R1Design and in silico evaluation of an mRNA vaccine against HTLV-1 using AI-driven reverse vaccinology approachesPLOS ONE

Dear Dr. Negahdaripour,

Thank you for submitting your manuscript to PLOS ONE. After careful consideration, we feel that it has merit but does not fully meet PLOS ONE’s publication criteria as it currently stands. Therefore, we invite you to submit a revised version of the manuscript that addresses the points raised during the review process.

We look forward to receiving your revised manuscript.

Kind regards,

Sheikh Arslan Sehgal, PhD

Academic Editor

PLOS ONE

Reviewers' comments:

Reviewer's Responses to Questions

**Comments to the Author**

1. If the authors have adequately addressed your comments raised in a previous round of review and you feel that this manuscript is now acceptable for publication, you may indicate that here to bypass the “Comments to the Author” section, enter your conflict of interest statement in the “Confidential to Editor” section, and submit your "Accept" recommendation.

Reviewer #1: All comments have been addressed

Reviewer #3: All comments have been addressed

2. Is the manuscript technically sound, and do the data support the conclusions?

Reviewer #1: Yes

Reviewer #3: Yes

3. Has the statistical analysis been performed appropriately and rigorously? 

Reviewer #1: N/A

Reviewer #3: N/A

4. Have the authors made all data underlying the findings in their manuscript fully available?

Reviewer #1: Yes

Reviewer #3: Yes

5. Is the manuscript presented in an intelligible fashion and written in standard English?

Reviewer #1: Yes

Reviewer #3: Yes

6. Review Comments to the Author

Reviewer #1: The authors have addressed all the comments appropriately. The manuscript can be recommended for publication.

Reviewer #3: PONE-D-25-00727R1

The manuscript was submitted by Seifi et al., has been described a reverse vaccinology reaching out to proper mRNA vaccine candidate against HTLV-1 infection with prophylactic method. However, the premise of the present manuscript is some novel, well written, and also might be considered as new approach to developing HTLV-1 vaccine candidate. Therefore, the authors must be addressed these issues completely, and then manuscript could be accepted for publication in PLOS one.

1- The authors have addressed to previous suggestions and comments as well.

2- Please explain HTLV-1 epidemiology in through the world briefly; particularly, in an endemic region of Iran with scholarly references.

3- The whole manuscript must be revised clinically and scientifically in terms of HTLV-1 infection. For example; “The virus is also involved in other clinical disorders, such as large granular lymphocytic (LGL) leukemia, HTLV-1-associated infectious dermatitis (IDH), uveitis, and other inflammatory signs”. Therefore, the sentences should be sound for reader. Alternatively, an expert human retrovirologist/ medical virologist colleague should be useful.

4- Please sort Keyword ascending. Please put the relevant reference(s) at the end of sentences “HTLV-1 transmission is mainly through the body's cell-containing fluids, including breast milk, blood, and semen. The virus usually exists in the intracellular form; however, its transmission is possible through direct contact”

5- In the introduction, for example; HTLV-1-infected patients must be divided in two categories. HTLV-1 asymptomatic carries (ACs) and/or HTLV-1 positive subjects.

6- Please apply “potential vaccine candidate” in the whole manuscript.

7- Far away, there is no vaccine candidate against HTLV-1 infection. Therefore, the authors have claimed mRNA vaccine strategy has prophylactic effects, while we need protective vaccine candidate against HTLV-1 infection worldwide. Therefore, the strongest strategy to HTLV-1/2 infection control and elimination is avoidance of spreading of HTLV-infection. Please explain this concern and include in the manuscript with acceptable scientific premise.

8- Please elaborate HTLV-1 pathogenesis with scholarly references shortly.

9- ACs has not needed any medication during their lifespan, please amend relevant sentences.

10- In spite of HTLV-1 is RNA virus, the virus could be integrate to human chromosome and persistent during lifespan, promote oncogenesis pathways or activate chronic inflammation responses. It seems that, mRNA vaccine strategy regarding HTLV-1/2 is not effective; moreover, might be disastrous. Consequently, researchers have been focused on the subunit vaccine against pathogens. Please explain and include in the manuscript with more explanation.

11- Please include HTLV-1 integrase/integrase inhibitor protein properties and its application for developing proper medication in the introduction and discussion sections.

12- In avoidance to Self-Citation, the authors must be removed some of previous self-cited studies.

13- Please include the limitations of current study, before conclusion section.

7. PLOS authors have the option to publish the peer review history of their article (what does this mean? ). If published, this will include your full peer review and any attached files.

**Do you want your identity to be public for this peer review?** For information about this choice, including consent withdrawal, please see our Privacy Policy .

Reviewer #1: No

Reviewer #3: No

---

## [Author Response · Author response to Decision Letter 2]

4 Dec 2025

Dear Editor,

Your comments and the reviewer's insights were highly insightful and enabled us to improve the quality of our manuscript greatly. The following pages include our amendments according to the journal requirements and our one-by-one responses to each of the reviewers' comments.

Revisions in the text are shown using track changes and highlights. Furthermore, the clear version of the revised manuscript (without track changes) was uploaded to the e-Component section.

1. The authors have addressed to previous suggestions and comments as well.

2. Please explain HTLV-1 epidemiology in through the world briefly; particularly, in an endemic region of Iran with scholarly references.

A brief explanation of the epidemiology and endemic areas of this virus was added to the introduction, as suggested. (page 3)

3. The whole manuscript must be revised clinically and scientifically in terms of HTLV-1 infection. For example; “The virus is also involved in other clinical disorders, such as large granular lymphocytic (LGL) leukemia, HTLV-1-associated infectious dermatitis (IDH), uveitis, and other inflammatory signs”. Therefore, the sentences should be sound for reader. Alternatively, an expert human retrovirologist/ medical virologist colleague should be useful.

We thank the reviewer for highlighting the need for clinical and scientific precision in describing HTLV-1-associated diseases. The sentence was revised to present clinical clarity. We have updated the manuscript to reflect these associations more accurately and have ensured that all clinical descriptions are consistent with current medical understanding. (page 3)

4. Please sort Keyword ascending. Please put the relevant reference(s) at the end of sentences “HTLV-1 transmission is mainly through the body's cell-containing fluids, including breast milk, blood, and semen. The virus usually exists in the intracellular form; however, its transmission is possible through direct contact”

The keywords have been alphabetically reordered as requested, and the specified reference has been added to the relevant section. (ref. 4)

5. In the introduction, for example; HTLV-1-infected patients must be divided in two categories. HTLV-1 asymptomatic carries (ACs) and/or HTLV-1 positive subjects.

As requested, the distinction between HTLV-1 asymptomatic carriers (ACs) and HTLV-1-positive subjects has been emphasized in the Introduction section by revising. (page 3)

6. Please apply “potential vaccine candidate” in the whole manuscript.

As requested, the phrase “potential vaccine candidate” has been applied consistently throughout the manuscript.

7. Far away, there is no vaccine candidate against HTLV-1 infection. Therefore, the authors have claimed mRNA vaccine strategy has prophylactic effects, while we need protective vaccine candidate against HTLV-1 infection worldwide. Therefore, the strongest strategy to HTLV-1/2 infection control and elimination is avoidance of spreading of HTLV-infection. Please explain this concern and include in the manuscript with acceptable scientific premise.

Thank you for your thoughtful feedback regarding clearly stating the position of immunization and other preventive approaches in the fight against HTLV-1. We added a paragraph to the end of the discussion section to clarify it. (pages 38 & 39) Additionally, a limitation section is included. Regarding our vaccine, the employed epitopes and predicted T-cell responses might qualify it as a candidate therapeutic vaccine too, but considering the lack of experience and vaccine in therapeutic setting, it needs a series of extensive additional evaluations and studies, so we did not focus on that.

8. Please elaborate HTLV-1 pathogenesis with scholarly references shortly.

A concise explanation of HTLV-1 pathogenesis has been added to the Introduction section and supported with scholarly references. (pages 3 & 4)

9. ACs has not needed any medication during their lifespan, please amend relevant sentences.

Relevant sentences have been added to clarify that asymptomatic carriers (ACs) typically do not require medication throughout their lifespan. (page 4)

10. In spite of HTLV-1 is RNA virus, the virus could be integrate to human chromosome and persistent during lifespan, promote oncogenesis pathways or activate chronic inflammation responses. It seems that, mRNA vaccine strategy regarding HTLV-1/2 is not effective; moreover, might be disastrous. Consequently, researchers have been focused on the subunit vaccine against pathogens. Please explain and include in the manuscript with more explanation.

We thank the reviewer for raising a critical concern regarding the suitability of mRNA vaccine platforms for HTLV-1. The question of whether mRNA vaccines are effective against HTLV-1 requires careful consideration. Importantly, mRNA vaccines do not integrate into host DNA and therefore do not share the pathogenic risks associated with retroviral integration. In principle, mRNA vaccines could elicit robust humoral and cellular immunity against HTLV-1 antigens, and the platform is well suited for inducing cytotoxic T-cell responses critical for controlling HTLV-1 infection. However, the primary challenge lies in the biology of HTLV-1 itself: because the virus persists mainly through clonal expansion of infected cells rather than ongoing virion production, vaccines must prevent initial infection rather than relying on post-exposure control. As a result, while mRNA-based approaches remain theoretically viable, current efforts continue to emphasize subunit vaccines due to their established safety profile, cost-effectiveness, and ease of antigen design within the constraints of limited HTLV-1 research funding. On the other hand, mRNA platforms offer distinct advantages in terms of safety, scalability, and rapid antigen delivery.

mRNA vaccines cannot integrate into the genome. Unlike HTLV-1, the synthetic mRNA used in vaccines:

-does not enter the nucleus

-does not undergo reverse transcription

-is rapidly degraded

-has no mechanism for integration

Therefore, mRNA vaccines do not pose a risk of contributing to HTLV-1–like integration events, and the mechanism of the virus does not make mRNA vaccination inherently dangerous. Importantly, the effectiveness of such vaccines will depend on their ability to generate sufficiently strong immune responses to block the earliest stages of infection, thereby preventing the establishment of proviral reservoirs that are otherwise resistant to immune clearance.

Accordingly, we have revised the Discussion section to include a concise clarification of the rationale for using an mRNA-based prophylactic strategy against HTLV-1. (page 35)

11. Please include HTLV-1 integrase/integrase inhibitor protein properties and its application for developing proper medication in the introduction and discussion sections.

The requested content has been carefully incorporated into both the Introduction and Discussion sections. (Pages 4 & 34-35)

12. In avoidance to Self-Citation, the authors must be removed some of previous self-cited studies.

Multiple self-citations with less importance were removed to improve the objectivity and coherence of our scientific manuscript. Considering our team’s experience in vaccine design, only a few very relevant previously published studies are kept to represent our background and experience in this field.

13. Please include the limitations of current study, before conclusion section.

We appreciate the reviewer’s suggestion and have incorporated a limitation section before the Conclusion.

I hope this modified version meets the high standards of PLOS One.

Thanks again for giving us this chance to improve our manuscript.

Best regards,

Manica Negahdaripour

Corresponding Author

---

## [Decision Letter · Decision Letter 2]

18 Dec 2025

Design and in silico evaluation of an mRNA vaccine against HTLV-1 using AI-driven reverse vaccinology approaches

PONE-D-25-00727R2

Dear Dr. Negahdaripour,

We’re pleased to inform you that your manuscript has been judged scientifically suitable for publication and will be formally accepted for publication once it meets all outstanding technical requirements.

Kind regards,

Sheikh Arslan Sehgal, PhD

Academic Editor

PLOS One

Additional Editor Comments (optional):

Reviewers' comments:

Reviewer's Responses to Questions

**Comments to the Author**

1. If the authors have adequately addressed your comments raised in a previous round of review and you feel that this manuscript is now acceptable for publication, you may indicate that here to bypass the “Comments to the Author” section, enter your conflict of interest statement in the “Confidential to Editor” section, and submit your "Accept" recommendation.

Reviewer #1: All comments have been addressed

Reviewer #3: All comments have been addressed

2. Is the manuscript technically sound, and do the data support the conclusions?

Reviewer #1: Yes

Reviewer #3: Yes

3. Has the statistical analysis been performed appropriately and rigorously? 

Reviewer #1: Yes

Reviewer #3: N/A

4. Have the authors made all data underlying the findings in their manuscript fully available?

Reviewer #1: Yes

Reviewer #3: Yes

5. Is the manuscript presented in an intelligible fashion and written in standard English?

Reviewer #1: Yes

Reviewer #3: Yes

6. Review Comments to the Author

Reviewer #1: The quality of the manuscript is improved as per suggestions. Hence, it can be accepted for publication.

Reviewer #3: The authors have been addressed the previous issues properly; therefor, current version of manuscript could be accepted for publication in PLOS One.

7. PLOS authors have the option to publish the peer review history of their article (what does this mean? ). If published, this will include your full peer review and any attached files.

**Do you want your identity to be public for this peer review?** For information about this choice, including consent withdrawal, please see our Privacy Policy .

Reviewer #1: No

Reviewer #3: No

---

## [Editor Report · Acceptance letter]

PONE-D-25-00727R2

PLOS One

Dear Dr. Negahdaripour,

I'm pleased to inform you that your manuscript has been deemed suitable for publication in PLOS One. Congratulations! Your manuscript is now being handed over to our production team.

Kind regards,

on behalf of

Dr Sheikh Arslan Sehgal

Academic Editor

PLOS One